# Filter Like You Test: Data-Driven Data Filtering for CLIP Pretraining

Mikey Shechter*             Yair Carmon*

## Abstract

We introduce Filter Like You Test (FLYT), an algorithm for curating large-scale vision-language datasets that *learns* the usefulness of each data point as a pretraining example. FLYT trains a scoring model that learns to weigh each example's features using gradient signals from downstream tasks training sets. Based on FLYT, we implement Mixing-FLYT (M-FLYT), which takes the per-example scores generated by different scoring methods as features, and learns to unify them into a single score. FLYT naturally produces a distribution over the training examples, which we leverage through Soft Cap Sampling (SCS), a strategy for obtaining a filtered pretraining dataset from per-example probabilities that samples examples while preventing over-representation through a repetition penalty. Using these methods, we achieve 40.1% ImageNet zero-shot accuracy on the DataComp medium scale filtering benchmark, a 2% absolute accuracy increase over all previous results and a 5.5% increase over results that—like us—use only public resources. Our approach also yields 37.7% on the average of 38 DataComp evaluation tasks, outperforming previous public-resource approaches by 0.4%.

## 1 Introduction

Filtering the pretraining data of CLIP models [35] has a dramatic effect on the resulting model's quality [39, 40, 17, 51]. Despite intense research and considerable progress, the best existing methods select data based on heuristic proxies for usefulness [16, 22, 49, 29, 27, 21]. In this work, we take a different approach: adhering to the core principle of machine learning, *we optimize a loss function that measures the usefulness of data for pretraining*.

We consider the setting of the DataComp filtering benchmark [17]. Our goal is to find a filtering method that selects a subset from a candidate pool of web-sourced image-text training data. To evaluate the quality of the selected subset, it is fed to a fixed training pipeline and the resulting model is evaluated on a fixed set of zero-shot image comprehension tasks. The DataComp rules do not constrain the computational cost of the filtering method. Moreover, DataComp allows using external data (except evaluation test sets) for making the filtering decisions, so long as the selected training set is a subset of the candidate pool.

In this setting, we introduce methods that together achieve state-of-the-art performance on the DataComp medium scale. At the core is *Filter Like You Test*[2] (FLYT), a method for training a scoring model that predicts the weight an image-text pair should have in CLIP pretraining. *Mixing-FLYT* (M-FLYT) is an implementation of FLYT that uses scores generated by existing scoring methods as input features to the scoring model. Complementing FLYT, *Soft Cap Sampling* (SCS) is a simple sampling strategy that leverages the probabilistic structure of FLYT's scores to construct the final pretraining dataset.

**Filter Like You Test (FLYT) (Section 3.2).** FLYT recasts data selection as data re-weighting [37], and learns to upweight (and hence select) the *pretraining* data that contributes the most to

---

*Tel Aviv University, `sachter@mail.tau.ac.il` and `ycarmon@tauex.tau.ac.il`.
[2]The name is inspired by FLYP [18].

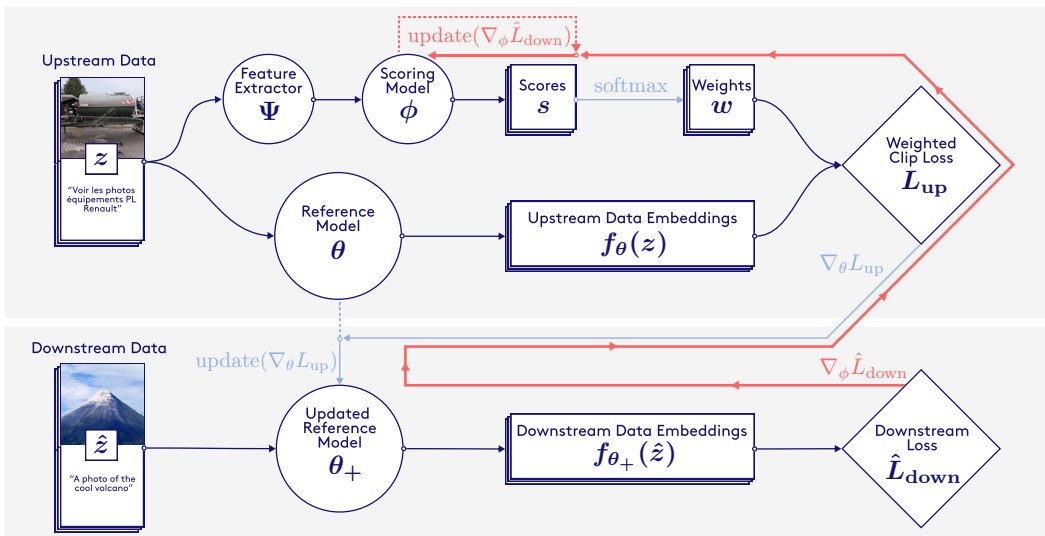

Figure 1: In each FLYT training loop, the scoring model takes features extracted from a batch of upstream data and generates a score for each example. These scores are converted to weights via softmax. A reference model processes the upstream batch, and the resulting embeddings, together with the weights, are used to compute a weighted CLIP loss. The reference model is then updated using gradients from this loss. Next, this updated reference model processes downstream data to compute a downstream loss, which produces a gradient signal that passes through the updated reference model all the way to the scoring model parameters.

the model's *downstream* performance. We jointly train two models: a *scoring model* that evaluates the usefulness of training examples using extracted features, and a *reference model* that provides feedback on these evaluations. During training, the scoring model assigns weights to each example in the batch, which are then used in our *weighted* CLIP loss function to update the reference model. The updated reference model is then evaluated on downstream data, allowing gradient signal from the downstream loss to flow through it, to the weights, and finally back to the scoring model. This allows the scoring model to give higher weights to examples that reduce the downstream loss of the reference model. We then use the trained scoring model to generate per-example scores. Figure 1 illustrates a single training step of FLYT and Algorithm 1 provides pseudocode.

**Mixing-FLYT (M-FLYT) (Section 3.4).** Previous work aggregates different filtering methods to obtain the best results. They either sum together different scores [22], take the intersection/union of the filtered datasets [17, 29, 22, 49] or pipeline the filtering methods one after the other [53, 49]. Departing from heuristic approaches, we leverage FLYT to directly optimize over a space of mixing methods defined by a simple linear combination of scores. Replacing the aggregation method of [22] with M-FLYT improves ImageNet accuracy by 1.3%. More importantly, M-FLYT can aggregate more scores. When used with 12 different scores, it outperforms the best single score by 1.7% on ImageNet accuracy, and by at least 0.8% on average accuracy. M-FLYT outperforms all baseline aggregation methods we evaluated, including summation and ImageNet accuracy-weighted sums by 0.5% on ImageNet accuracy and 0.6% on average accuracy.

A different natural choice for FLYT's feature extractor component is a multimodal neural network, such as CLIP; we refer to this Embeddings-FLYT (E-FLYT). While E-FLYT does not improve the state-of-the-art on its own, incorporating its scores as an input to M-FLYT leads to improved overall performance.

**Soft Cap Sampling (SCS) (Section 3.6).** FLYT produces a probability distribution over the training examples, allowing us to obtain a filtered subset by simply sampling from the distribution. However, sampling with replacement produces too much repetition of the top-scoring examples, and sampling without replacement produces too few. To interpolate between the two sampling methods we propose SCS, a strategy that samples with replacement but reduces repetition using a penalty term (see Algorithm 2). Using only publicly available resources, M-FLYT with SCS has 40.1% ImageNet zero-shot accuracy, which is a 2% absolute increase compared to the best prior work, and a 5.5%

| **Algorithm 1:** Filter Like You Test (FLYT) | **Algorithm 2:** Soft Cap Sampling (SCS) |
|---|---|

**Algorithm 1:** Filter Like You Test (FLYT)

**Input:** Scoring model $q_{\phi_0}$ with initial parameters $\phi_0$, reference model $f_{\theta_0}$ with initial parameters $\theta_0$, feature extractor $\Psi$, upstream dataset $D$, downstream dataset $C$

**Parameters:** Number of steps $T$, upstream data batch size $B$, downstream data batch size $B'$, weighted loss $\ell_{\text{WCL}}$, and downstream loss $\ell_{\text{DS}}$

**for** $t = 0$ **to** $T - 1$ **do**
    $z_{1:B} \sim D,\ \hat{z}_{1:B'} \sim C$
    $\zeta_{1:B} = \Psi(z_{1:B})$
    $s_{1:B} = q_{\phi_t}(\zeta_{1:B})$
    $w_{1:B} = \text{softmax}(s_{1:B})$
    $L_{\text{up}} = \ell_{\text{WCL}}(f_{\theta_t}(z_{1:B}), w_{1:B})$
    $\theta_{t+1} = \text{update}(\nabla_{\theta_t} L_{\text{up}})$
    $\hat{L}_{\text{down}} = \ell_{\text{DS}}(f_{\theta_{t+1}}(\hat{z}_{1:B'}))$
    $\phi_{t+1} = \text{update}(\nabla_{\phi_t} \hat{L}_{\text{down}})$
    ▷ Appendix A

**return** $\phi_T$

**Algorithm 2:** Soft Cap Sampling (SCS)

**Input:** Training examples $z_{1:M}$ and their scores $s_{1:M}$

**Parameters:** Repetition penalty $\alpha$, target dataset size $N$ and sample count per iteration $G$

$\hat{D} = [\ ]$
**while** $|\hat{D}| < N$ **do**
    $w_{1:M} = \text{softmax}(s_{1:M})$
    $i_{1:G} = $ Sample $G$ indices according to distribution $w_{1:M}$ without repetitions.
    $\hat{D} = \text{concatenate}(\hat{D}, z_{i_{1:G}})$
    $s_{i_{1:G}} = s_{i_{1:G}} - \alpha$

**return** $\hat{D}$

improvement compared to the best prior work that uses only publicly available resources. Our filtered dataset also achieves 37.7% average accuracy across all evaluation tasks, which is within 0.1% of our reproduction of the best previously published filtering method, and a 0.4% improvement over the best prior work using only publicly available resources. On the DataComp small scale benchmark (which we did not evaluate until the end of the project) our method outperforms previous work on both ImageNet and average accuracy (see Table 4).

**Data and code release.** We share all the code required to reproduce our work, including E-FLYT, M-FLYT training, and SCS sampling at `https://github.com/formll/FLYT`. Additionally, we publish our best-performing scoring and mixing models along with the resulting filtered datasets, and the M-FLYT input scores at `https://huggingface.co/collections/formll/flyt-67bb167366ec0fa0d5b8e4bd`.

## 2  Related work

Early work on curating large-scale image-text datasets established methods to extract, filter, and organize pairs of images and captions from the web [41, 5, 45, 13]. Beyond assembling large collections, several lines of research address dataset pruning and deduplication [44, 2], example characterization [28, 47], and quality assessment [34].

CLIP [35] leveraged large image-text datasets via contrastive learning, producing the first generation of vision-language foundation models and revolutionizing computer vision. While the original CLIP was trained on a proprietary set of 400M image-text pairs, subsequent efforts introduced open-source alternatives. LAION-400M [39] and LAION-5B [40] filtered large web-crawled corpora primarily using the cosine similarity between image and text embeddings from a pretrained CLIP model, along with basic image and text quality metrics. Xu et al. [51] attempted to reproduce CLIP's original data pipeline and managed to outperform the original model.

Recognizing the importance of data quality for CLIP training, Gadre et al. [17] introduces the DataComp benchmark. DataComp fixes the data pool as well as the training and evaluation procedures, and allows researchers to focus on comparing filtering methods while keeping all other parameters fixed. Below, we survey the literature on data filtering for DataComp.

**Filtering methods.** Gadre et al. [17] consider several baseline filtering methods, and achieved best results by combining CLIP score with a clustering-based filtering method using similarity to the ImageNet training set. T-MARS [29] observe that images containing texts that overlap with the

caption received high CLIP scores, while not contributing much to the model training. To address this issue, they mask out these texts before calculating CLIP score. For better results, they take the intersection of this approach and their variation of Maini et al. [28]. Yu et al. [53] developed a pipeline merging various techniques categorized as: single modality filtering, cross modality filtering, and distribution alignment. DFN [16] use CLIP score from a model fine-tuned on a large high-quality private data. HYPE [22] argues CLIP's image-text alignment alone is insufficient, and adds individual information of the images and texts. They use hyperbolic embeddings to generate a hyperbolic similarity score, and image and text specificity scores, which they sum with the traditional Euclidean CLIP and the image-based scores. s-CLIPLoss [49] introduce a variation to the standard CLIP score that normalizes it by the example's similarity score with other examples in a batch, and after filtering using this score, filters remaining data using a downstream example similarity metric. While these heuristic approaches are successful, our approach directly learn how to estimate an example's quality.

**Data sampling.** Given a scoring of the data pool, most filtering approaches select the top 15-20% of examples, resulting in each example appearing 5–7 times during training in the DataComp benchmark. Goyal et al. [19] analyzed the quality-quantity trade-off of dataset filtering, and found that for the compute budget of the medium scale DataComp benchmark selecting the top 20% of examples is optimal, with more aggressive filtering leading to diminishing returns. However, their analysis only considers uniform sampling while top performing methods created non-uniformly sampled datasets through concatenation of multiple subsets of data [22, 49]. Our proposed SCS method extends this approach of non-uniform sampling using a simple probabilistic procedure; in our top performing dataset, individual examples appear up to 56 times.

**Learned data weighting.** FLYT shares conceptual and technical similarities with meta-learning paradigms. Particularly similar meta-learning works address the problem of learned data weighting. Ren et al. [37] introduced an algorithm for online example weighting which utilizes gradient signals from a validation dataset for producing per-example weights. Shu et al. [43] employs a similar idea to Ren et al. [37] with the addition of a weighting model that is optimized to weigh the examples. FLYT is similar to these methods in that it too learns weights and backpropogates through a "target loss" after a gradient update on the "training loss." However, unlike FLYT, these methods consider online data weighting for standard supervised learning, with the intention of improving robustness to noise and data imbalance, mostly in small-scale settings.

**Concurrent work.** Concurrent with our research, two related works also developed data filtering algorithms evaluated on the DataComp benchmark. Xu et al. [52] introduced EcoDatum which transforms unimodal and multimodal scores into discrete labels and uses a weak supervision ensemble model to aggregate them based on agreements and disagreements [36]. Their approach is similar to M-FLYT in that it combines multiple filtering methods, but unlike us, it doesn't optimize a downstream objective for aggregation. Separately, Engstrom et al. [14] proposed metagradient descent (MGD), a model-free dataset curation method that, similar to our approach, leverages gradient signals from downstream tasks. While MGD learns to score each example separately, we train a scoring model that applies to unseen examples, which allows us to extend our method to larger dataset scales at a low computational cost. Additionally, whereas MGD optimizes for all available downstream tasks in the DataComp benchmark, we use only the ImageNet training set as the downstream data. MGD sets a new state-of-the-art average accuracy of 40.2% on the DataComp medium benchmark, though its ImageNet accuracy is only 27%.

## 3 Method

In this section, we describe FLYT in detail. We introduce our notation by going over standard CLIP training (Section 3.1), then move to describing FLYT (Section 3.2). We then go over the weighted CLIP loss we use to train FLYT (Section 3.3), followed by two different feature extractors for FLYT's implementation (Section 3.4), and our choice of downstream data (Section 3.5). Finally, we present SCS (Section 3.6).

### 3.1 Notation and problem setting

**CLIP training.** We denote a CLIP training example $z = (z^I, z^T) \in D$ where $D$ is our pretraining dataset and $z^I, z^T$ are the training image and text, and denote a batch of $B$ training examples by $z_{1:B}$. We train a CLIP model with parameters $\theta$ that maps each example $z$ to features $f_\theta(z) = (f_\theta(z^I), f_\theta(z^T))$. Letting $f_\theta(z_{1:B})$ denote the features of all the examples in the batch, the standard

CLIP loss is

$$\ell_{\mathrm{C}}\left(f_\theta(z_{1:B})\right) = (\ell_{\mathrm{image}} + \ell_{\mathrm{text}})/2, \tag{1}$$

$$\text{where} \quad \ell_{\mathrm{image}} = \ell_{\mathrm{contrast}}(f_\theta(z^I), f_\theta(z^T)), \quad \ell_{\mathrm{text}} = \ell_{\mathrm{contrast}}(f_\theta(z^T), f_\theta(z^I)),$$

$$\ell_{\mathrm{contrast}}(u_{1:B}, v_{1:B}) = \sum_{i=1}^{B} -\log\left(\frac{e^{\tau\langle u_i, v_i\rangle}}{\sum_{j=1}^{B} e^{\tau\langle u_i, v_j\rangle}}\right),$$

and $\tau$ is the learnable temperature parameter.

## 3.2 Filter Like You Test (FLYT)

In this section we describe FLYT. Our goal is to learn a *scoring model* $q_\phi$ parameterized by $\phi$. The scoring model takes a feature vector $\zeta = \Psi(z)$ produced by applying feature extractor $\Psi$ to example $z$, and maps it to a score $s$, such that the higher $s$ is, the more useful $z$ is as a pretraining example. We overload notation to let

$$s_{1:B} = q_\phi(\zeta_{1:B})$$

denote the scores computed element wise over the batch. We obtain weights over the batch $w_{1:B}$ using a softmax transformation:

$$w_{1:B} = \mathrm{softmax}(s_{1:B}) \text{ where } \mathrm{softmax}(s_i) = \frac{\exp(s_i)}{\sum_{j=1}^{B} \exp(s_j)}.$$

Now consider a *reference* CLIP model $f_\theta$. We define a *weighted* CLIP loss

$$L_{\mathrm{up}} = \ell_{\mathrm{WCL}}(f_\theta(z_{1:B}), w_{1:B})$$

which is a function of both the reference model embeddings, and the weights produced by the scoring model. This loss serves as an approximation to the loss we would have gotten had we used the scoring model to filter the batch. We discuss the weighted CLIP loss in detail in section Section 3.3.

We can use this loss and its gradient to update $\theta$; let

$$\theta_+(w_{1:B}) = \mathrm{update}(\nabla_\theta L_{\mathrm{up}})$$

be the result of a gradient-based update to $\theta$. For example, using SGD with learning rate $\eta$, we have $\theta_+(w_{1:B}) = \theta - \eta\nabla_\theta L_{\mathrm{up}}$. Using a more sophisticated optimizer like AdamW, $\theta_+(w_{1:B})$ takes a more complicated form, but we can still think about it as a function of $w_{1:B}$.

Next, we introduce the key part of our approach, a batch of *downstream* data (e.g., the ImageNet training set). Let $\hat{z}_{1:B'}$ denote this batch (note that $B'$ does not necessarily have to be equal to $B$). We use this batch to obtain a gradient signal for updating $w_{1:B}$ and hence the scoring model parameters $\phi$. To that end, define:

$$\hat{L}_{\mathrm{down}} = \ell_{\mathrm{DS}}(f_{\theta_+(w_{1:B})}(\hat{z}_{1:B'})), \tag{2}$$

Where $\ell_{\mathrm{DS}}$ is some loss over the downstream data: we experiment with standard CLIP loss $\ell_{\mathrm{C}}$, as well as cross-entropy (CE), with or without temperature; see Appendix C.8 for details. We can then update $\phi$ using $\nabla_\phi \hat{L}_{\mathrm{down}}$.

Finally, we score each example with our trained scoring model, then filter based on these scores.

## 3.3 Weighted CLIP loss

Our definition of weighted CLIP loss follows these guiding principles:

1. Crucially, a larger weight means the corresponding example is more represented in the loss.
2. Setting an example's weight to 0 should be equivalent to excluding it from the batch.
3. Uniform weights should result in the standard CLIP loss.
4. The weight should reflect the example's importance both as a positive and as a negative example.

To realize these principle we modify the CLIP loss Equation (1), showing modifications in red:

$$\ell_{\mathrm{WCL}}\left(f_\theta(z_{1:B}), w_{1:B}\right) = \frac{\ell_{\mathrm{image}}^{\mathrm{w}} + \ell_{\mathrm{text}}^{\mathrm{w}}}{2}, \tag{3}$$

where

$$\ell_{\text{image}}^{\text{w}} = \ell_{\text{contrast}}^{\text{w}} \left( f_\theta(z^I), f_\theta(z^T), w_{1:B} \right), \quad \ell_{\text{text}}^{\text{w}} = \ell_{\text{contrast}}^{\text{w}} \left( f_\theta(z^T), f_\theta(z^I), w_{1:B} \right), \text{and}$$

$$\ell_{\text{contrast}}^{\text{w}} \left( u_{1:B}, v_{1:B}, w_{1:B} \right) = \sum_{i=1}^{B} -w_i \log \left( \frac{w_i e^{\tau \langle u_i, v_i \rangle}}{\sum_{j=1}^{B} w_j e^{\tau \langle u_i, v_j \rangle}} \right).$$

### 3.4 Feature extractors

**Mixing-FLYT (M-FLYT)**   Inspired by past success in mixing multiple filtering heuristics [17, 22, 49, 53], we implement FLYT using existing scoring methods as our feature extractor. Specifically, the feature extractor $\Psi$ produces a vector $\zeta \in \mathbb{R}^k$ of $k$ different scores, generated by $k$ different scoring methods. Our scoring model $q_\phi : \mathbb{R}^k \to \mathbb{R}$ then takes as input the different scores to output a single unified score $s$. We refer to this implementation of a scoring model as a *mixing model*.

Using M-FLYT gives us the advantage of not needing to learn example importance from scratch, and instead use the information gained by other scoring methods. In fact, we notice that a simple linear mixing model is enough for obtaining strong results.

**Embedding-FLYT (E-FLYT)**   With E-FLYT, we use a pretrained CLIP model as the feature extractor $\Psi$ to generate image and text embeddings. Our scoring model $q_\phi$ then maps the concatenated image and text embeddings to a score $s$.

Despite CLIP's ability to produce semantically rich representations, our current E-FLYT implementation does not yet match M-FLYT's performance. Appendix C.7 documents our experiments, which show improvements but suggest further work is needed to fully leverage these embeddings.

### 3.5 Choice of downstream data

FLYT requires high-quality downstream data to effectively learn which examples are more valuable for pretraining. For our implementation, we use the ImageNet training set as a proxy for downstream task data, which aligns with the DataComp competition rules permitting "use of other images associated with these tasks (e.g., supervised training sets)" [17]. This approach follows standard practice, as all other top-performing methods [17, 49, 53, 16] similarly utilize the ImageNet training set for their filtering algorithms. The DataComp benchmark includes 38 diverse tasks, some of which differ significantly from ImageNet. This range of evaluation tasks means that the average downstream score provides a practical measure of model performance on unfamiliar tasks.

### 3.6 Soft Cap Sampling (SCS)

We now consider the task of obtaining a filtered pretraining dataset from per-example scores. A principled data filtering approach should use these scores for deciding the number of repetitions of each example in the filtered dataset. It is not clear how to do that when using a similarity based score as in most prior work. In contrast, a sampling strategy emerges directly from the definition of FLYT, since the scores naturally correspond to the (log) probabilities of including examples in the batch.

A straightforward approach would be to sample from this distribution 128M times to create our final dataset. In practice, doing that results in over-represented examples. To correct this, we introduce SCS (Soft Cap Sampling): every time we add an example to our filtered data, we subtract a "repetition penalty" $\alpha$ from its score, reducing the probability it is sampled again. For implementation efficiency, we perform the sampling procedure iteratively. In each iteration, we sample a batch of $G = 100K$ examples without repetitions, then subtract $\alpha$ from the score of each batch member. Algorithm 2 presents SCS, and Figure 2 illustrates the distribution generated via SCS.

## 4 Experiments

In this section, we present the steps we took to create our top performing dataset which achieves 40.1% zero-shot ImageNet accuracy on the DataComp medium scale benchmark and 37.7% average accuracy across all DataComp evaluation tasks. We first review key details about our training and evaluation setup (Section 4.1), then discuss the challenges, design choices and results for M-FLYT (Section 4.2) and SCS (Section 4.3). Table 1 summarizes our findings and compares them to the state-of-the-art. In Section 4.4 we apply our algorithm to the DataComp small scale benchmark, demonstrating that our approach works effectively across different data scales.

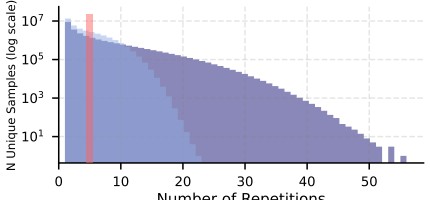

Figure 2: Histogram of example repetitions using SCS on the probabilities produced by M-FLYT. See Section 4.3 for more details.

Table 1: Comparison of our method to top performing methods on the DataComp medium scale filtering benchmark. DFN-FT is our best reproduction of DFN [16]. ∗ denotes results from the original paper, and + indicates evaluations using our downloaded 119M candidate pool using the authors' provided UIDs. Table 15 compares our evaluations to the results reported in the corresponding papers.

| Filtering Method | Public Resources | ImageNet | ImageNet dist. shifts | VTAB | Retrieval | Average |
|---|---|---|---|---|---|---|
| Image-based∩CLIP score∗ [17] | Yes | 0.305 | 0.243 | 0.342 | 0.250 | 0.328 |
| DFN-FT [our reproduction of 16] | Yes | 0.342 | 0.274 | 0.357 | 0.290 | 0.348 |
| HYPE 20%∗ [22] | Yes | 0.338 | 0.269 | 0.357 | 0.286 | 0.343 |
| s-CLIPLoss+ [49] | Yes | 0.333 | 0.273 | 0.361 | 0.251 | 0.352 |
| DFN+ [16] | No | 0.367 | 0.303 | 0.371 | 0.280 | 0.364 |
| HYPE+DFN+ [22] | No | 0.372 | 0.304 | 0.374 | 0.287 | 0.368 |
| DFN+s-CLIPLoss+ [49] | No | 0.371 | 0.304 | 0.390 | 0.285 | **0.378** |
| HYPE+DFN+s-CLIPLoss+ [49] | No | 0.381 | 0.310 | 0.392 | 0.284 | **0.378** |
| M-FLYT | Yes | 0.359 | 0.294 | 0.383 | **0.310** | 0.371 |
| M-FLYT+SCS | Yes | **0.401** | **0.311** | **0.396** | 0.292 | 0.377 |

## 4.1 General setup

We experiment mainly with the medium scale DataComp filtering track benchmark [17]. In this benchmark, the goal is to filter a 128M example candidate pool sourced from Common Crawl [1]. The filtered data is then used to train a ViT-B/32 CLIP using a fixed procedure, and the resulting model undergoes a fixed evaluation on 38 downstream tasks, with performance typically reported in four main categories: ImageNet, ImageNet distribution shifts, VTAB, and retrieval, along with the overall average across all tasks.

For our experiments we were only able to download 119M out of the 128M DataComp candidate pool. To compare our filtered datasets to those published in prior work, we reapply the DataComp training and evaluation procedure on the intersection of the published datasets and our downloaded pools. Table 15 shows that the accuracy degradation due to the missing data is between 0.1% and 1%.

The fixed DataComp medium training procedure goes through 128M input examples (hence using a fixed amount of compute) regardless of the size of the filtered data; if the filtered dataset has less than 128M examples, it is reshuffled and repeated until reaching 128M examples. In all experiments except those specifically evaluating sampling strategies, we follow the standard strategy of selecting the top 20% scoring examples, which means each example is seen by the model 5 or 6 times during training. Using SCS, we control how many times each example will be seen, forming a dataset with repeating data and using a shuffle buffer to spread data repetitions across training.

For both M-FLYT (Section 4.2) and E-FLYT (Appendix C.7), we use a CLIP ViT-B/32 model as the reference model. Unless otherwise mentioned, we use the unfiltered DataComp medium pool as the upstream dataset, and the ImageNet training set as the downstream dataset.

To enable large-scale training of FLYT, we develope a data parallelism approach, which is detailed in Appendix A. The computational costs are outlined in Appendix C.2.

Table 2: Left: Performance and linear weight of each input to M-FLYT. For HYPE [22] and s-CLIPLoss [49], we replicate the scores and mixing methodology used in the original papers. Right: Baseline aggregation experiments as described in Section 4.2.

| Input | Weight | ImageNet | Average |
|---|---|---|---|
| ViT-B/32 | 0.08 | 0.282 | 0.327 |
| ViT-L/14 | 0.21 | 0.267 | 0.317 |
| DFN-Base | 0.60 | 0.297 | 0.333 |
| DFN-FT | **0.80** | **0.342** | 0.348 |
| HYPE $-d_L$ | 0.45 | | |
| HYPE $\epsilon_i$ | -0.05 | 0.309 | 0.341 |
| HYPE $\epsilon_t$ | 0.02 | | |
| s-CLIPLoss | 0.51 | 0.331 | **0.363** |
| $\mathrm{NormSim}_\infty$ | 0.55 | | |
| IN1K-Classifier | 0.36 | 0.268 | 0.294 |
| CC2M-Classifier | 0.17 | 0.287 | 0.308 |
| E-FLYT | 0.65 | 0.316 | 0.323 |

| Baseline | ImageNet | Average |
|---|---|---|
| Sum | 0.327 | 0.331 |
| Standardized Sum | 0.348 | 0.360 |
| IN-weighted $r = 2$ | 0.350 | 0.364 |
| IN-weighted $r = 4$ | 0.354 | 0.359 |
| IN-weighted $r = 8$ | 0.353 | 0.365 |
| IN-weighted $r = 16$ | 0.352 | 0.362 |
| M-FLYT | **0.359** | **0.371** |

## 4.2 M-FLYT results

We train a linear mixing model using input scores that are standardized to zero mean and unit standard deviation. The standardization both improves performance and provides interpretable weights for each input method. We empirically found that initializing the reference model with a CLIP model trained using DataComp's medium scale configuration on their unfiltered dataset and using standard cross entropy (CE) as the downstream loss produced better results; see Appendix C.3

We create and reproduce 12 input scores. We use the CLIP score of OpenAI's ViT-B/32 and ViT-L/14 models [35], as well as two reproduced DFNs [16], before and after fine-tuning, which we refer to as DFN-Base and DFN-FT (Appendix B.1). We reproduce the image and text specificity $\epsilon_i, \epsilon_t$, and negative Lorentzian distance $-d_L$ introduced by HYPE [22] using their codebase, and the s-CLIPLoss and $\mathrm{NormSim}_\infty$(IN1K) scores introduced by Wang et al. [49] using theirs. We also add three new inputs. Two binary classifiers that classify between some "high quality" dataset and the DataComp medium scale dataset, and an E-FLYT scoring model described in Appendix C.7. See Appendix B for more details about the inputs.

We compare M-FLYT to its individual input components and to several baseline approaches. These baselines include a simple sum of all input scores and a sum of scores after standardization to zero mean and unit standard deviation. Finding that standardization significantly improves performance, we further explore weighted standardized sums where inputs are weighted by their normalized ImageNet scores adjusted to achieve a specific max-to-min weight ratio $r$, for details see Appendix C.5.

Table 2 demonstrate that M-FLYT outperforms all individual scores as well as all baseline aggregations. Table 9 demonstrates that excluding any individual input score from M-FLYT's training leads to performance degradation, suggesting that the method effectively leverages multiple signals, and could potentially achieve stronger performance given more (or better) input scores.

In Appendix E we complement our quantitative results with a qualitative comparison of filtering methods. In particular, we plot the top-scoring training example according to different filtering methods. Figures 5 to 7 show that M-FLYT prioritizes examples containing multiple ImageNet classes, while sorting by OpenAI's ViT-B/32 CLIP scores favors images with visible text that matches the caption text.

## 4.3 SCS results

For SCS, we experiment with a repetition penalty $\alpha$ between 0.1 and 0.6, and sample in batches of $G = 100K$. We choose $G$ to balance efficient sampling while ensuring the resulting dataset closely approximates what we would get using $G = 1$. Appendix C.10 shows that the exact value of $G$ has minimal impact on performance. When preparing the filtered dataset, we use a 1.5M shuffle buffer in the DataComp resharder to prevent batches from containing many copies of the same example.

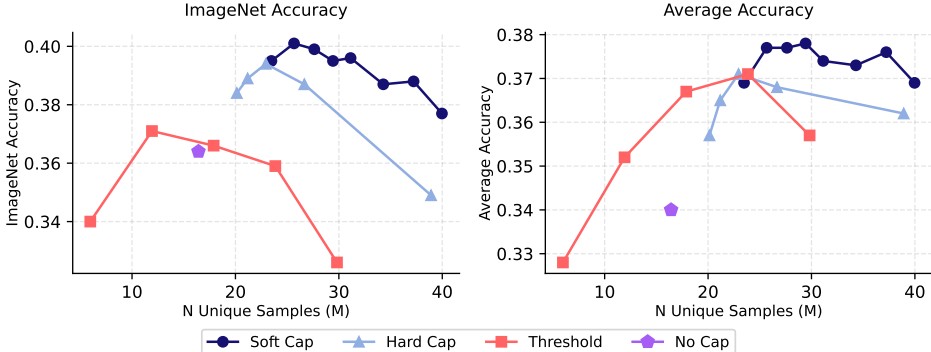

Figure 3: Comparison of sampling strategies (SCS, HCS, threshold filtering, and No Cap) showing their effect on ImageNet accuracy (left) and average accuracy (right). SCS was tested with $\alpha$ values from 0.1 to 0.6, HCS with $\beta$ values from 5 to 25 and standard threshold with values from 5% to 25%

Table 3: Using SCS with M-FLYT compared to IN-weighted.

| Mixing method | Sampling method | ImageNet | Average |
|---|---|---|---|
| M-FLYT | Threshold top 20% | 0.359 | 0.371 |
| M-FLYT | SCS ($\alpha = 0.15$) | **0.401** | **0.377** |
| IN-weighted | Threshold top 20% | 0.353 | 0.365 |
| IN-weighted | SCS ($\alpha = 0.15$) | 0.379 | 0.362 |

When using the scores produced by M-FLYT, SCS outperforms every other filtering method in the DataComp ImageNet Leaderboard (Table 1). Figure 3 shows the influence of the repetition penalty parameter $\alpha$, and compares SCS to standard threshold filtering, regular sampling (No Cap), as well as to a sampling method we call Hard Cap Sampling (HCS). HCS allows an example to be sampled until it reaches a "frequency limit" $\beta$, and then its sampling probability is set to 0. We note that while SCS outperforms HCS, both methods outperform the standard threshold filtering.

Figure 3 also shows a pattern where for threshold filtering, example diversity improves average accuracy but reduces ImageNet accuracy: selecting the top 10% of examples (11.93M) performs better on ImageNet compared to selecting the top 20% (23.86M), but results in lower average accuracy across tasks. This trade-off is not as apparent with SCS.

To test SCS independently of M-FLYT, we apply it with a repetition penalty $\alpha = 0.15$ to scores from our strongest baseline, IN-weighted with $r = 8$. We convert these scores to probabilities via softmax, following the same approach as with M-FLYT. Table 3 shows that SCS improves IN-weighted ImageNet accuracy by 2.6% while decreasing average score by 0.3%. In comparison, applying SCS to M-FLYT improves ImageNet accuracy by 4.2% and average score by 0.6%. This demonstrates that a key advantage of FLYT is that it naturally produces a distribution over training examples, which can be leveraged through sampling approaches like SCS.

### 4.4 Small scale results

We evaluate our method on the small scale DataComp benchmark. We use the mixing model trained by M-FLYT described in Section 4.2, and sample using SCS with a repetition penalty $\alpha = 0.5$ and a batch size $G = 10K$. Table 4 shows that our method achieves 8% zero-shot ImageNet zero-shot accuracy and 18.5% average accuracy, outperforming previous work on both categories. These results suggest that M-FLYT and SCS can be effectively used across different computational scales. Figure 4 shows the effect of different $\alpha$ values.

## 5 Limitations and future work

Computational scale is a clear limitation of our work: we only had the resources to experiment at the DataComp medium scale, and could not test our methods on the larger scales. However, in prior work

Table 4: Comparing our methods to top performing methods on the DataComp small scale filtering benchmark. ∗ denotes results from the DataComp leaderboard that did not provide a paper to cite.

| Filtering method | ImageNet | Average |
|---|---|---|
| EcoDatum [52] | 0.053 | 0.182 |
| WS [21] | 0.056 | 0.180 |
| HYPE [22] | 0.063 | 0.180 |
| CAM ∩ Flipped CLIP∗ | 0.064 | 0.178 |
| M-FLYT | 0.072 | **0.185** |
| M-FLYT + SCS ($\alpha = 0.5$) | **0.080** | **0.185** |

progress on medium scale usually translated to progress on larger scales, which makes us optimistic that our proposed methods will also scale well.

The computational overhead of FLYT is a potential limitation, but may also be negligible in some scaling regimes. In our experiments, M-FLYT achieves high performance using approximately $3\times$ fewer FLOPs than training a DataComp medium-scale model, due to training on relatively few examples (20M). Scaling up the number of training steps would proportionally increase computational costs, as detailed in Appendix C.2. Whether the scoring model needs to increase in computational cost as the data and final model size increases—and hence the relative overhead of FLYT in large scale settings—remains a topic for future work.

Another notable limitation is that E-FLYT underperforms methods such as DFN, though its scores did contribute to M-FLYT. To surpass state-of-the-art methods we had to leverage existing filtering methods through M-FLYT rather than relying on a general feature extractor. We believe this is likely a deficiency in our optimization stack, solvable by more careful tuning of learning rate (schedules) for individual components and regularization techniques such as dropout. In similar vein, we believe that more careful parameter tuning should allow us to leverage downstream data that is more diverse than ImageNet, though these hypotheses require empirical validation.

A fourth limitation—and direction for future research—is that we use data weighting as a proxy for data selection when defining the FLYT objective function. It would be interesting to try to model data selection more directly via reinforcement learning: at each reference model training step, the scoring model picks a batch of training data and observes a reward signal for the chosen data.

A fifth direction for future work is closing the data selection / pretraining loop by iteratively re-training the scoring model to select the next data to pretrain our model on. FLYT is particularly well-suited to this form of active learning, as it allows us to use the pretraining model as the reference model at each scoring iteration, ensuring each selected batch is optimal for the model's current state.

Finally, FLYT should be useful past DataComp and vision-language data filtering. In particular, it will be exciting to extend it to data filtering and mixing for language models.

## Acknowledgments

We thank Sonia Lichtenzveig Sela and Alexander Tsverin from the TAU CS systems team for crucial technical support. We thank our anonymous reviewers for experiment and writing suggestions, Alex Fang and Vaishaal Shankar for helpful discussion of the DFN implementation details, and Sanghyuk Chun for sharing the HYPE codebase.

This research was partially supported by the computing resources of the Tel Aviv University Center for AI and Data Science, the Adelis Foundation, and the Israeli Science Foundation (ISF) grant no. 2486/21.

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

# A  Implementing data-parallel distribution

CLIP training benefits from a large batch size [35, 7]. FLYT trains a CLIP model while weighting each example compared to others in the batch, which also requires a sufficiently large batch size. However, training FLYT requires more memory than training a standard CLIP model since we have to keep the computational graph from $\phi$ to $\hat{L}_{\text{down}}$ in order to calculate $\nabla_\phi \hat{L}_{\text{down}}$. This means that in order to train FLYT on a large scale, we need to implement data-parallel distributed training.

Our goal is to compute $\nabla_\phi \hat{L}_{\text{down}}$ using data parallelism across multiple GPUs, which is not trivial considering FLYT introduces different dependencies between examples in the batch. We start by using the chain rule to write:

$$\nabla_\phi \hat{L}_{\text{down}} = \left[ \frac{\partial w_{1:B}}{\partial \phi} \right] \nabla_{w_{1:B}} \hat{L}_{\text{down}},$$

and focus the second factor

$$\nabla_{w_{1:B}} \hat{L}_{\text{down}} = \nabla_{w_{1:B}} \ell_{\text{DS}}(f_{\theta_+(w_{1:B})}(\hat{z}_{1:B'})),$$

where

$$\theta_+(w_{1:B}) = \text{update}(g(w_{1:B}))$$

with

$$g(w_{1:B}) = \nabla_\theta \ell_{\text{WCL}}(f_\theta(z_{1:B}), w_{1:B})$$

and $\text{update}(\cdot)$ is an optimizer update function. We then use the chain rule again to write:

$$\nabla_{w_{1:B}} \hat{L}_{\text{down}} = \left[ \frac{\partial g}{\partial w_{1:B}} \right] \nabla_g \ell_{\text{DS}}(f_{\text{update}(g)}(\hat{z}_{1:B'})),$$

We may further apply the chain rule (and the fact the partial derivatives commute) to write

$$g = \sum_{i \in [B]} \frac{\partial \ell_{\text{WCL}}(f_{1:B}, w_{1:B})}{\partial f_i} \nabla_\theta f_\theta(z_i)$$

where $f_i$ is shorthand for the embedding $f_\theta(z_i)$ that implicitly says we only need to store the value of the embedding and not the computational graph behind it. This way each GPU calculates $g$ for its own sub-batch, and we can then gather and sum their result to get $g$ for the full batch using the CLIP gradient accumulation technique from Cherti, Mehdi et al. [7]. Using this, we define

$$v \triangleq \nabla_g \ell_{\text{DS}}(f_{\text{update}(g)}(\hat{z}_{1:B'})),$$

which we can again compute using gradient accumulation. Treating $v$ as a constant vector independent of $w_{1:B}$, the gradient we would like to calculate is

$$\nabla_{w_{1:B}} \hat{L}_{\text{down}} = \frac{\partial \langle g, v \rangle}{\partial w_{1:B}} = \frac{\partial}{\partial w_{1:B}} \sum_{i \in [B]} \frac{\partial \ell_{\text{WCL}}(f_{1:B}, w_{1:B})}{\partial f_i} \langle \nabla_\theta f_\theta(z_i), v \rangle.$$

If we define the function

$$h_i(\eta) = \frac{\partial \ell_{\text{WCL}}(f_{1:B}, w_{1:B})}{\partial f_i} f_{\theta + \eta v}(z_i)$$

Then our gradient is simply

$$\nabla_{w_{1:B}} \hat{L}_{\text{down}} = \sum_{i \in [B]} \frac{\partial}{\partial w_{1:B}} h_i'(0).$$

Which is amenable to gradient accumulation.

The first factor

$$\frac{\partial}{\partial \phi} w_{1:B} = \frac{\partial}{\partial \phi} \text{softmax}(q_\phi(\zeta_{1:B}))$$

can be calculated separately using gradient accumulation, and we get

$$\nabla_\phi \hat{L}_{\text{down}} = \left[ \frac{\partial w_{1:B}}{\partial \phi} \right] \nabla_{w_{1:B}} \hat{L}_{\text{down}}.$$

Table 5: DFN-Base reproduction experiments.

| Initialization | Num. samples | Notes | Training dataset | ImageNet | Average |
|---|---|---|---|---|---|
| Random | 10M | | CC12M | 0.230 | 0.300 |
| Random | 50M | | CC12M | 0.274 | 0.324 |
| Random | 128M | | CC12M | 0.281 | 0.328 |
| OpenAI | 128M | DFN-Base | CC12M | **0.297** | 0.333 |
| OpenAI | 128M | | CC12M+CC3M+SS15M | 0.286 | 0.334 |
| OpenAI | 512M | 2x Batch Size (8192) | CC12M | 0.296 | **0.338** |

Table 6: DFN-FT reproduction experiments

| Initialization | Num. samples | Notes | Training dataset | ImageNet | Average |
|---|---|---|---|---|---|
| DFN-Base | 400K | DFN-FT | IN1K+MS COCO+Flickr30k | **0.342** | **0.348** |
| DFN-Base | 1.6M | | IN1K+MS COCO+Flickr30k | 0.340 | **0.348** |
| DFN-Base | 3M | | IN1K+MS COCO+Flickr30k | 0.335 | 0.338 |
| DFN-Base | 400K | 1/10 LR (5e-5) | IN1K+MS COCO+Flickr30k | 0.321 | 0.344 |
| DFN-Base | 1M | 1/10 LR (5e-5) | IN1K+MS COCO+Flickr30k | 0.324 | 0.342 |
| DFN-Base | 12M | 1/10 LR (5e-5) | IN1K+MS COCO+Flickr30k | 0.319 | 0.340 |
| DFN-Base | 800K | | IN1K | 0.333 | 0.343 |

## B    M-FLYT input details

In order to experiment with M-FLYT, we reproduce some existing scoring methods and create some new ones:

1. **CLIP scores.** We use the CLIP scores calculated using OpenAI ViT-B/32 and ViT-L/14 CLIP models [35].

2. **DFN.** The results of Fang et al. [16] are proprietary and we do not have access to their best DFN model or the scores created by it. The authors provided a version that uses only publicly available resources, but it was not trained using all the improved techniques they discovered. Thus, we reproduced their work using only publicly available resources. We use the CLIP score of 2 DFN networks: one before fine-tuning on ImageNet, and one after. Appendix B.1 provides the details of our reproduction.

3. **HYPE.** Kim et al. [22] provided a PyTorch implementation which includes their hyperbolic CLIP weights, and the functionality required to reproduce the image specificity $\epsilon_i$, text specificity $\epsilon_t$, and negative Lorentzian distance $-d_L$. We used these three values as inputs.

4. **s-CLIPLoss.** Wang et al. [49] also provided a PyTorch implementation. We used OpenAI ViT-B/32 embeddings to calculate s-CLIPLoss and $\text{NormSim}_\infty(\text{IN1K})$.

5. **Binary classifiers.** We trained two linear binary models that classify between a "high quality" dataset and the DataComp medium scale dataset. We describe them in Appendix B.2 below.

6. **E-FLYT.** We use our E-FLYT-Base scoring model, as detailed in Appendix C.7.

### B.1   DFN reproduction

Since M-FLYT requires good inputs to perform well, we made effort to train a high performing DFN. In the paper, Fang et al. [16] trained a ViT-B/32 CLIP model initialized with OpenAI's checkpoint on the high-quality private dataset HQITP-350M. Since we do not have access to this dataset, we trained our DFN on CC12M [41]. Similar to Fang et al. [16], we then fine-tune on the combined training sets of IN1K with sampled OpenAI templates as texts, MS COCO, and Flickr30k. Unless otherwise specified, we use the same hyper-parameters as DataComp medium scale training. Tables 5 and 6 show results we obtained with different hyperparameter choices when our reproductions DFN-Base and DFN-FT, respectively.

Table 7: Detailed hyper parameters of the different configurations used for training and ablations of M-FLYT and E-FLYT. ∗ Initialization "DataComp medium" means we train a model from scratch on the unfiltered DataComp medium scale dataset using the DataComp medium scale training configuration.

| Parameter | E-FLYT-Base | E-FLYT-22 | E-FLYT-LR | M-FLYT-Base |
|---|---|---|---|---|
| Optimizer | AdamW | AdamW | AdamW | AdamW |
| Weight decay | 0.2 | 0.2 | 0.2 | 0.2 |
| $(\beta_1, \beta_2)$ | $(0.9, 0.98)$ | $(0.9, 0.98)$ | $(0.9, 0.98)$ | $(0.9, 0.98)$ |
| Scheduler | cosine | cosine | cosine | cosine |
| Reference model LR | 5e-5 | 5e-5 | Variable | 5e-5 |
| Scoring model LR | 1e-3 | 1e-3 | Variable | 1e-3 |
| Feature extractor init | OpenAI | OpenAI | OpenAI | None |
| Reference model init | OpenAI | OpenAI | OpenAI | DataComp medium$^*$ |
| Downstream loss | CE + temperature | CLIP | CE | CE |
| Downstream batch size $B'$ | 3072 | 768 | 1536 | 3072 |
| Upstream batch size $B$ | 4096 | 4096 | 4096 | 4096 |
| Num. downstream samples | 15.4M | 3.8M | 7.7M | 15.4M |
| Num. upstream samples | 20.5M | 20.5M | 20.5M | 20.5M |
| Num. steps | 5000 | 5000 | 5000 | 5000 |
| Num. warmup steps | 100 | 100 | 100 | 100 |
| Downstream dataset | ImageNet | 22 tasks | ImageNet | ImageNet |
| Upstream dataset | DataComp medium | DataComp medium | DataComp medium | DataComp medium |
| Trainable reference parameters | all | all | all | all |
| Trainable scoring parameters | all | all | all | all |

## B.2 Binary classifiers input

When training our top performing M-FLYT, we use two binary linear classifiers. These models take as input embeddings generated by OpenAI's ViT-B/32 model and predict which of two datasets these embeddings come from. IN1K-Classifier classifies between the ImageNet training set and the Data-Comp medium scale dataset image embeddings. CC2M-Classifier classifies between concatenated image and text embeddings of the DataComp medium scale dataset and *CC2M*. CC2M is a dataset of 2M examples we created by running our IN1K-Classifier on the CC12M dataset [41] and taking the top 2M examples that the model classified as ImageNet examples. As shown in Table 2, while IN1K-Classifier underperforms compared to CLIP score filtering, it still demonstrates improvement over no filtering and receives substantial weighting from M-FLYT. In contrast, CC2M-Classifier achieves performance similar to CLIP score filtering but receives a lower M-FLYT weighting. We trained them for 40K steps with a batch size of 256, a cosine learning rate with a maximal value of 0.01 and 500 warm-up steps.

# C  Experimental setup

## C.1  Experiment configurations

When training M-FLYT and E-FLYT, we optimize the mixing model, the embeddings scoring model, and the reference models using AdamW [26] with a weight decay of 0.2, $(\beta_1, \beta_2) = (0.9, 0.98)$, and a cosine scheduler [25]. Due to computational constraints, some experiments compare against different model configurations. We list their full hyper parameters in Table 7 and describe their main differences here. We refer to the configuration from Section 4.2 as *M-FLYT-Base*, and to the E-FLYT configuration used in the experiments described in Appendix C.7 as *E-FLYT-Base*. *E-FLYT-22* maintains the same parameters as E-FLYT-Base, with three key differences: it employs the standard CLIP loss as the downstream loss, uses a downstream batch size $B'$ of 768, and trains with 22 downstream tasks. *E-FLYT-LR* uses the same configuration as E-FLYT-Base, except it employs the CE loss without temperature and uses a downstream batch size $B'$ of 1536; this configuration was used exclusively for learning rate experiments.

## C.2  Computational cost

In terms of FLOPs, M-FLYT requires slightly more than $2\times$ the FLOPs of regular CLIP model training per example, while E-FLYT requires slightly more than $3\times$. Since we only train our models

Table 8: Experiments using the M-FLYT-Base configuration (Table 7).

| Experiment | Baseline | Variation | ImageNet | Average |
|---|---|---|---|---|
| M-FLYT | — | — | 0.359 | 0.371 |
| Downstream loss | CE | CE + temperature | 0.352 | 0.366 |
| Reference model initialization | Trained on DC medium | Random initialization | 0.341 | 0.354 |
| | | OpenAI | 0.342 | 0.355 |
| Input standardization | Yes | No | 0.323 | 0.337 |

Table 9: Comparison of training M-FLYT with all input methods vs leaving one method out. Numbers in parentheses show performance degradation compared to using all inputs. We highlight values with the largest degradation, indicating which methods provide the most benefit when included.

| Inputs | ImageNet | ImageNet dist. shifts | VTAB | Retrieval | Average |
|---|---|---|---|---|---|
| All Inputs | 0.359 | 0.294 | 0.383 | 0.310 | 0.371 |
| No ViT-B/32, ViT-L/14 | 0.355 (0.004) | 0.288 (_0.006_) | 0.367 (_0.016_) | 0.307 (0.003) | 0.360 (0.011) |
| No DFN-Base, DFN-FT | 0.341 (**0.018**) | 0.280 (**0.014**) | 0.375 (0.008) | 0.281 (**0.029**) | 0.358 (_0.013_) |
| No $-d_L, \epsilon_i, \epsilon_t$ | 0.351 (_0.008_) | 0.291 (0.003) | 0.366 (**0.017**) | 0.308 (0.002) | 0.364 (0.007) |
| No s-CLIPLoss, NormSim$_\infty$ (IN1K) | 0.354 (0.005) | 0.292 (0.002) | 0.377 (0.006) | 0.307 (0.003) | 0.364 (0.007) |
| No IN1K-Classifier, CC2M-Classifier | 0.352 (0.007) | 0.288 (_0.006_) | 0.368 (0.015) | 0.301 (_0.009_) | 0.355 (**0.016**) |
| No E-FLYT | 0.352 (0.007) | 0.292 (0.002) | 0.376 (0.007) | 0.309 (0.001) | 0.366 (0.005) |

for 20M examples (compared to DataComp's 128M), M-FLYT uses approximately $3\times$ fewer FLOPs than training a DataComp medium scale model, and E-FLYT uses approximately $2\times$ fewer.

With our implementation, M-FLYT takes 40 A100 hours to train, and E-FLYT takes 45 A100 hours, roughly equivalent to running a DataComp medium scale experiment. For SCS with a batch size $G = 100K$, it takes about 100 minutes to sample 128M examples using a naive Python implementation on a single CPU. For larger scales (DataComp large or Xlarge), the filtering computation becomes negligible relative to the overall training cost.

Comparing to other filtering methods:

- Fang et al. [16] reported (in Table 2) training their ViT-B/32 DFN for 5.12B samples seen, which requires approximately 80-120x more FLOPs than our method

- Kim et al. [22] trained a ViT-L/14 CLIP model on 8 V100 GPUs for 256M samples, taking 488 GPU hours (61 hours $\times$ 8 GPUs)

- Wang et al. [49], which doesn't require additional model training beyond the pretrained OpenAI CLIP model, reported 5 hours of preprocessing on an L40 machine

### C.3  M-FLYT ablations

Table 8 shows experiments using M-FLYT. The standard CE loss slightly outperforms CE with temperature (Appendix C.8), and reference model initialized with "DataComp medium" (explained in Appendix C.7) outperforms OpenAI initialization. We also find that input standardization greatly increases performance.

Table 9 shows the result of training M-FLYT while excluding groups of related inputs. Removing any input group degrades performance, with several notable results: DFN reproduction scores have the strongest impact on ImageNet, ImageNet distribution shifts, and retrieval scores (degrading performance by 1.8%, 1.4%, and 2.9% respectively). For VTAB tasks, excluding HYPE scores causes the largest drop (1.7%), while unexpectedly, removing the linear classifiers—despite being the weakest standalone filtering methods—leads to the largest decline in average performance (1.6%).

Table 10: Comparing M-FLYT to the original mixing method of HYPE [22] and s-CLIPLoss [49]. HYPE combines scores by summing their three components ($-d_L + \epsilon_i + \epsilon_t$), while s-CLIPLoss applies a two-stage filtering process: selecting the top 30% based on s-CLIPLoss scores, then selecting the top 20% according to their NormSim scores from this filtered subset.

| Input method | Mixing method | ImageNet | Average |
|---|---|---|---|
| HYPE | Original (sum scores) | 0.309 | 0.341 |
|  | M-FLYT | 0.322 | 0.347 |
| s-CLIPLoss and NormSim | Original (two stage) | 0.331 | 0.363 |
|  | M-FLYT | 0.334 | 0.357 |

## C.4 Comparing M-FLYT to aggregation methods in prior work

Table 10 compares M-FLYT against the original mixing strategies of HYPE and s-CLIPLoss. For HYPE's components, M-FLYT outperforms the simple summation approach on both ImageNet (+1.3%) and average (+0.6%) metrics. When mixing s-CLIPLoss scores, M-FLYT performs similarly to the original two-stage filtering method, showing marginal improvement on ImageNet (0.3%) but slightly lower average performance (0.6%).

## C.5 M-FLYT baseline - ImageNet-weighted standardized sum

Here we explain the ImageNet-weighted standardized sum baseline we compare against in Section 4.2. For $K$ ImageNet accuracies $\text{IN}_{1:K}$ corresponding to $K$ scoring methods, we first standardize the per-example scores produced by these methods individually across all training examples to have zero mean and unit standard deviation. We denote by $s_i^j$ the standardized score that method $i \in 1, ..., K$ gave example $z_j$.

We then normalized the ImageNet accuracies $\text{IN}_{1:K}$ between zero and one, and add a scalar to achieve our desired ratio $r$:

$$\hat{w}_i = \frac{\text{IN}_i - \min(\text{IN}_{1:K})}{\max(\text{IN}_{1:K}) - \min(\text{IN}_{1:K})},$$

$$w_i = \hat{w}_i + \frac{1}{r - 1},$$

where $\min(\text{IN}_{1:K})$ is the lowest ImageNet score, and $\max(\text{IN}_{1:K})$ is the highest. This ensures that the ratio between the maximum and minimum weights equals $r$:

$$r = \frac{\max(w_{1:K})}{\min(w_{1:K})}.$$

The final score for example $z_j$ is then $\sum_{i=1}^{K} w_i s_i^j$.

In Table 2 we experiment with max-to-min ratios $r \in \{2, 4, 8, 16\}$.

## C.6 E-FLYT learning rate tuning

Table 11 compares different scoring model and reference model learning rates using the E-FLYT-LR configuration. We notice that E-FLYT is not very sensitive to the scoring model LR (with reference LR fixed at 1e-4) and select 1e-3 based on slightly better results. E-FLYT is somewhat sensitive to reference model LR, and with scoring model LR fixed at 1e-3, 5e-5 gives the best results. These values are used for subsequent experiments.

## C.7 E-FLYT results

For E-FLYT, we use a CLIP ViT-B/32 [35] as the feature extractor and an $\text{FFN}_{\text{GLU}}$ [42] scoring model. We initialize both the feature extractor and the reference model with OpenAI's checkpoint.

In Table 12 we show experiments using E-FLYT-Base. We evaluate three alternatives for the downstream loss function: The CLIP loss, as defined in Equation (1) $\ell_{\text{C}}(f_{\theta_+(w_{1:B})}(\hat{z}_{1:B'}))$, standard cross entropy (CE) loss, and CE with a temperature parameter (see definitions in Appendix C.8). Our experiments show that CLIP loss and CE with temperature demonstrate similar performance, both outperforming the standard CE loss. For subsequent experiments, we use CE with temperature

Table 11: Learning rate tuning using the E-FLYT-LR configuration (Table 7). We first optimize the scoring model LR with reference model LR fixed at 1e-4, then optimize reference model LR with scoring LR fixed at 1e-3.

| Scoring model LR | Reference model LR | ImageNet | Average |
|---|---|---|---|
| 5e-4 | 1e-4 | 0.286 | 0.300 |
| 1e-3 | 1e-4 | 0.286 | 0.306 |
| 5e-3 | 1e-4 | 0.280 | 0.310 |
| 1e-2 | 1e-4 | 0.277 | 0.300 |
| 1e-3 | 1e-5 | 0.158 | 0.238 |
| 1e-3 | 5e-5 | **0.292** | **0.319** |
| 1e-3 | 1e-4 | 0.286 | 0.306 |
| 1e-3 | 5e-4 | 0.267 | 0.296 |

because it better resembles the accuracy metric used to evaluate most DataComp tasks. We found it essential to use separate learned temperature parameter for the upstream and downstream loss functions when using either the CLIP or CE with temperature loss. We initialize the downstream temperature parameter $\tau_{\text{DS}}$ to 1/0.07, as is the default when training a CLIP model from scratch, while the initial value for the upstream temperature is set to 100 as part of the pretrained reference model weights. The downstream temperature is updated during training using $\nabla_{\tau_{\text{DS}}} \hat{L}_{\text{down}}$.

Table 12 also compares reference model and feature extractor initializations, where initializing with stronger models outperform weaker ones. We evaluate four initializations in ascending order of performance: Random initialization, "DataComp medium", where we train a model from scratch on the unfiltered DataComp medium scale dataset using the DataComp medium scale training configuration, OpenAI checkpoint, and "DataComp-13B", which is a model trained by Gadre et al. [17] on the DC-1B dataset for 13B samples seen. Further results in the table show that training the reference model is much better than freezing it, and that fine-tuning the feature extractor prevents the training from succeeding. Our ablation studies show that performance remains stable with downstream batch sizes ranging from 1024 to 4096. Similarly, performance is consistent when training for 1250 to 12500 steps, suggesting that E-FLYT could be optimized using less compute.

As discussed in Section 5, we believe that our E-FLYT training stack has significant room for improvement.

### C.8 Downstream loss

We now explain how construct the downstream loss from a downstream batch $\hat{z}_{1:B'}$ containing labeled data of the form $\hat{z}_i = (x_i, c_i)$, where $x_i$ is an image and $c_i \in \{1, \ldots, K\}$ is a class label. First, we uniformly sample templates for each of the classes following DataComp's class-specific template definitions $t_{1:K} = \text{sample}(1:K)$. Then, the CE loss is given by

$$\ell_{\text{DS}}(\hat{z}_{1:B'}) = \sum_{i=1}^{B'} -\log\left(\frac{e^{\tau_{\text{DS}}\langle f_\theta(x_i), f_\theta(t_{c_i})\rangle}}{\sum_{j=1}^{K} e^{\tau_{\text{DS}}\langle f_\theta(x_i), f_\theta(t_j)\rangle}}\right),$$

where $\tau_{\text{DS}}$ is a learnable temperature parameter we only use in the CE+temperature variation for the downstream data. Using the standard CE loss, $\tau_{\text{DS}}$ is set to 1. Using this notation, we define the downstream CLIP loss mentioned in Appendix C.7 with a separate temperature parameter $\tau_{\text{DS}}$ as

$$\ell_{\text{C}}(\hat{z}_{1:B'}) = \sum_{i=1}^{B'} -\log\left(\frac{e^{\tau_{\text{DS}}\langle f_\theta(x_i), f_\theta(t_{c_i})\rangle}}{\sum_{j=1}^{B'} e^{\tau_{\text{DS}}\langle f_\theta(x_i), f_\theta(t_{c_j})\rangle}}\right) + \sum_{i=1}^{B'} -\log\left(\frac{e^{\tau_{\text{DS}}\langle f_\theta(t_{c_i}), f_\theta(x_i)\rangle}}{\sum_{j=1}^{B'} e^{\tau_{\text{DS}}\langle f_\theta(t_{c_i}), f_\theta(x_j)\rangle}}\right).$$

### C.9 E-FLYT with diverse downstream data

Instead of using the ImageNet training set alone for the downstream data, we experiment with using 22 training sets from the DataComp benchmark. We now detail the training configuration, and then discuss the results of experiments using E-FLYT-22. First, we list the datasets used:

Table 12: Experiments using the E-FLYT-Base configuration (Table 7). ∗ Initialization "Trained on DC-1B" means using the ViT-B-32 model trained by Gadre et al. [17] on DC-1B for 13B samples seen (Hugging Face model card).

| Experiment | Baseline | Variation | ImageNet | Average |
|---|---|---|---|---|
| E-FLYT-Base | — | — | 0.316 | 0.323 |
| Downstream loss | CE + temperature | CE (no temperature) | 0.287 | 0.305 |
| | | CLIP | 0.313 | 0.330 |
| Reference model initialization | OpenAI | Random initialization | 0.270 | 0.312 |
| | | Trained on DC medium | 0.290 | 0.308 |
| | | Trained on DC-1B∗ | 0.314 | 0.320 |
| Feature extractor initialization | OpenAI | Random initialization | 0.157 | 0.234 |
| | | Trained on DC medium | 0.287 | 0.302 |
| | | Trained on DC-1B∗ | 0.325 | 0.329 |
| Reference model + feature extractor initialization | OpenAI | Trained on DC-1B∗ | 0.326 | 0.331 |
| Trainable parameters | Scoring + reference model | Only scoring model | 0.251 | 0.298 |
| | | + Feature extractor | 0.135 | 0.227 |
| Num. training steps | 5000 | 12500 | 0.306 | 0.320 |
| | | 1250 | 0.311 | 0.329 |
| Downstream Batch Size | 3072 | 1024 | 0.315 | 0.313 |
| | | 4096 | 0.311 | 0.316 |

- ImageNet [12]
- MNIST [11]
- SVHN [31]
- CIFAR10 [24]
- CIFAR100 [24]
- Food-101 [4]
- Oxford Flowers-102 [32]
- Oxford-IIIT Pet [33]
- iWildCam [3]
- SUN-397 [50]
- EuroSAT [20]
- Stanford Cars [23]
- DTD [9]
- RESISC45 [6]
- GTSRB [46]
- FGVC Aircraft [30]
- Pascal VOC 2007 [15]
- Country211 [35, 48]
- Rendered SST2 [54]
- FMoW [8]

Table 13: Different downstream tasks experiments using the E-FLYT-22 configuration (Table 7).

| Downstream tasks variation | ImageNet | Average |
|---|---|---|
| Imbalanced 22 tasks | 0.315 | 0.330 |
| Balanced 22 tasks | 0.274 | 0.311 |
| Only ImageNet | 0.313 | 0.330 |

Table 14: SCS parameter ablations: Effects of repetition penalties and batch sizes.

| Repetition penalty $\alpha$ | Batch size $G$ | ImageNet | Average |
|---|---|---|---|
| 0.10 | 100K | 0.395 | 0.369 |
| 0.15 | 100K | **0.401** | 0.377 |
| 0.20 | 100K | 0.399 | 0.377 |
| 0.25 | 100K | 0.395 | **0.378** |
| 0.30 | 100K | 0.396 | 0.374 |
| 0.40 | 100K | 0.387 | 0.373 |
| 0.50 | 100K | 0.388 | 0.376 |
| 0.60 | 100K | 0.377 | 0.369 |
| 0.15 | 10K | 0.399 | 0.373 |
| 0.15 | 1M | 0.398 | 0.371 |

- Dollar Street [38]

- STL-10 [10]

When preparing our datasets, we split them into tar files of 100 examples each in webdataset format. During training, we use a shuffle buffer of 50,000 examples. Since the webdataset loader reads entire tar files at once, small shards and a large shuffle buffer help ensure diverse tasks within each training batch. The combined dataset contains 2M examples, with ImageNet making up the majority. In Table 13, we compare different dataset configurations. Using the combined dataset without balancing classes yields performance very similar to using only the ImageNet dataset. When we balance the datasets to ensure equal representation (where the model sees samples from each downstream dataset the same number of times), performance deteriorates. This performance drop might be due to some tasks containing as few as 1020 total training samples, potentially causing overrepresentation of these smaller datasets during training.

## C.10   SCS ablations

With SCS, we evaluate different repetition penalties and find that performance varies smoothly with $\alpha$, reaching optimal results around $\alpha = 0.15$ (Table 14). For sampling efficiency, we introduce the batch size $G$, setting it to $100K$. This value is large enough for efficient medium scale sampling yet sufficiently small based on our empirical observation that while each sample had 1280 ($128M/100K$) opportunities to be selected, our most frequently sampled example appears only 56 times. To more directly validate our choice of $G$, Table 14 shows that using values of $G$ of different order of magnitude has little effect on performance.

## C.11   DataComp small scale results

For the small scale DataComp benchmark, we reuse the mixing model from the medium scale without retraining E-FLYT or M-FLYT. The only changes were re-tuning the repetition penalty $\alpha$ for the smaller dataset and reducing the sampling batch size $G$ by a factor of 10 to $10K$, aligning with the dataset size reduction. Figure 4 shows that the smaller scale performs best with a higher repetition penalty, around $\alpha = 0.5$, compared to the medium scale which performs better with values between 0.15 to 0.25. A higher repetition penalty reduces the number of repetitions per example, which becomes more important for smaller datasets.

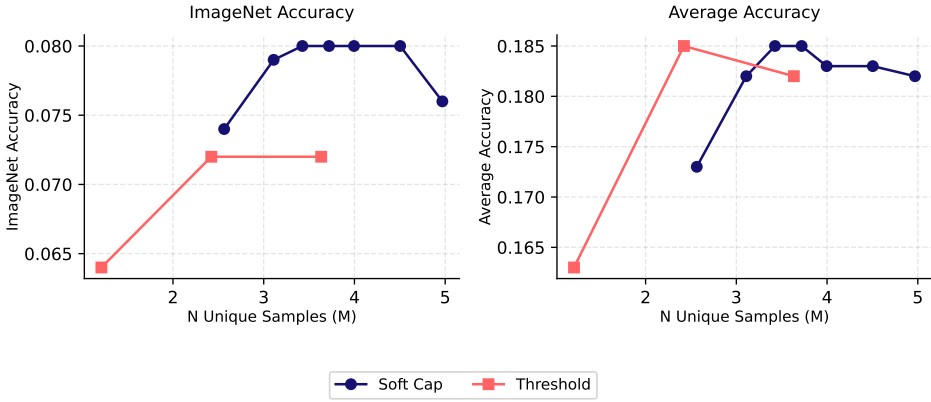

Figure 4: Comparing SCS to hard threshold filtering on the small scale DataComp benchmark. SCS was tested with $\alpha$ values from 0.15 to 1, and standard threshold with values from 10% to 30%

Table 15: Comparison between the results reported in the original papers and our reproduction using their provided UIDs. Some experiments were run using the complete DataComp medium scale dataset (128M) while our reproduction used the 119M examples we were able to download. Results marked with $*$ were performed by Wang et al. [49] on their 110M downloaded dataset; they also report partial results using the full 128M data pool. Our best results are listed below.

| Filtering Method | Pool Size | ImageNet | ImageNet dist. shifts | VTAB | Retrieval | Average |
|---|---|---|---|---|---|---|
| s-CLIPLoss | 110M | $0.324^*$ | $0.274^*$ | $0.359^*$ | $0.263^*$ | $0.352^*$ |
| | 119M (Reprod.) | 0.333 | 0.273 | 0.361 | 0.251 | 0.352 |
| DFN | 128M | 0.371 | 0.298 | 0.388 | 0.288 | 0.373 |
| | 119M (Reprod.) | 0.367 | 0.303 | 0.371 | 0.280 | 0.364 |
| HYPE+DFN | 116M | 0.382 | 0.303 | 0.393 | 0.306 | 0.379 |
| | 119M (Reprod.) | 0.372 | 0.304 | 0.374 | 0.287 | 0.368 |
| DFN+s-CLIPLoss | 128M, $110M^*$ | 0.375 | $0.309^*$ | $0.386^*$ | $0.281^*$ | 0.386 |
| | 119M (Reprod.) | 0.371 | 0.304 | 0.390 | 0.285 | 0.378 |
| HYPE+DFN+s-CLIPLoss | 128M, $110M^*$ | 0.382 | $0.314^*$ | $0.385^*$ | $0.276^*$ | 0.388 |
| | 119M (Reprod.) | 0.381 | 0.310 | 0.392 | 0.284 | 0.378 |
| M-FLYT+SCS ($\alpha = 0.15$) | 119M | 0.401 | 0.311 | 0.396 | 0.292 | 0.377 |

## D Reproduction comparison

Table 15 compares model we train using publicly available DataComp subsets to the numbers reported in the papers that published them. The results are generally similar, but not identical, with accuracy difference of up to 1%. The main driver of the difference appears to be the size of the actual pool used to train the model: we were able to download only 119M of the 128M examples in the DataComp medium scale pool, while the authors of the other papers had access to a more complete subset of the examples.

## E Qualitative analysis

Figures 5 to 7 show the top-scoring examples for M-FLYT, standard CLIP score, and DFN-FT. We observe that M-FLYT prioritizes examples containing multiple ImageNet classes. For instance, the highest-scoring example features the caption "Honey the Golden Retriever stands in a tent," where both "Golden Retriever" and "tent" correspond to ImageNet classes. In contrast, the CLIP score filtering method using OpenAI's ViT-B/32 model favors images with visible text that matches the caption text.

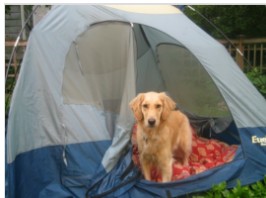

Honey the Golden Retriever stands in a tent.

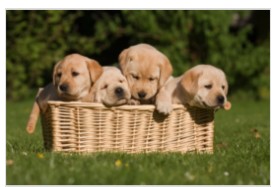

four Labrador puppies sitting in a basket photo

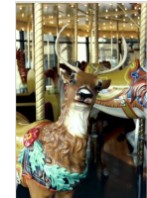

Many carousel deer, like this Herschell-Spillman one at the 1928 carousel at Van Andel Museum Center in Grand Rapids, Michigan, have real antlers. (© Jean Bennett, via carousels.org)

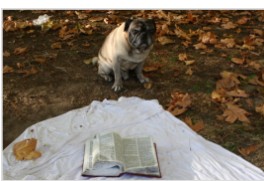

Pug and Abandoned Bible on Sheet

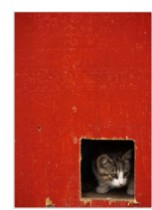

A kitten peers out of a square hole in a red farm building

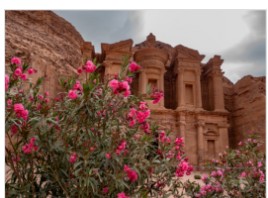

Pink wild desert flowers growing in front of the monastery at Petra, Jordan

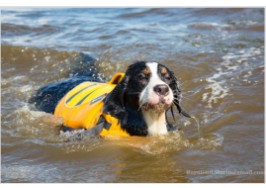

Bernese Mountain Dog Swimming with her Ruffwear Lifejacket, K9 Float Coat in Lake Pepin on the Mississippi River in Lake City, MN

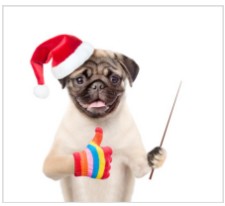

christmas hat: Pug puppy in red christmas hat holding a pointing stick and showing thumbs up. isolated on white background. Stock Photo

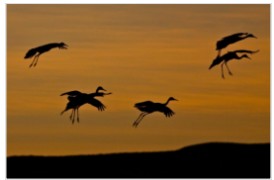

Sandhill Cranes fly in at sunset.

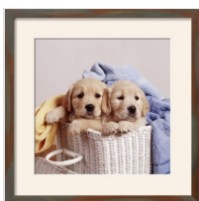

'Golden Retriever Dog Two Puppies in Laundry Basket' Framed Photographic Print Frame: Grayson Mahogany/Brown Framed, Size: 23" H x 23" W 15204308

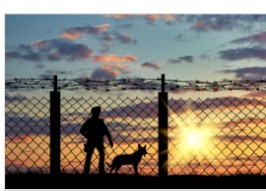

prison: Silhouette of a soldier on the border with a fence and a dog at sunset Stock Photo

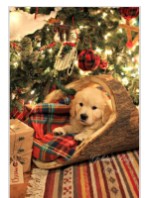

RUG!!!! Golden Retriever puppy under Christmas tree in log basket with plaid blanket - www.goldenboysandme.com

Figure 5: M-FLYT top scoring examples.

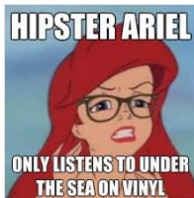

Vinyl Meme - hipster ariel only
listens to under the sea on vinyl
hipster

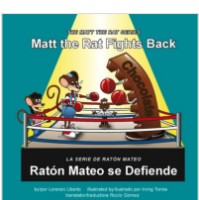

Download Matt The Rat Fights Back /
Raton Mateo Se Defiende (The Matt
the Rat Series / La Serie de Ratón
Mateo) (English and Spanish
Edition) ebook

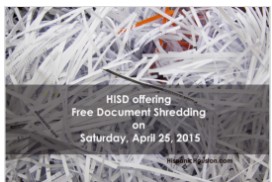

HISD Free Document Shredding on
Saturday, April 25, 2015

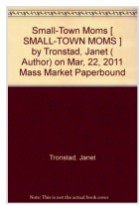

Download Small-Town Moms [ SMALL-
TOWN MOMS ] by Tronstad, Janet (
Author) on Mar, 22, 2011 Mass
Market Paperbound PDF

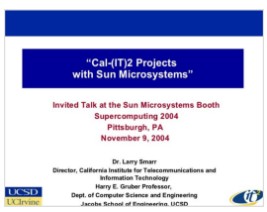

" Cal-(IT)2 Projects  with Sun
Microsystems" Invited Talk at the
Sun Microsystems Booth
Supercomputing 2004 Pittsburgh,
PA...

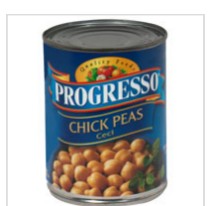

Progresso Chick Peas 19oz Can
product image

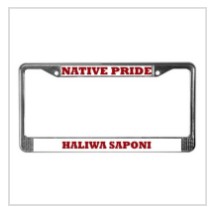

Native Pride Haliwa Saponi License
Plate Frame

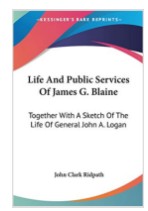

Life And Public Services Of James
G. Blaine: Together With A Sketch
Of The Life Of General John A.
Logan (0548490120) by Ridpath, John
Clark

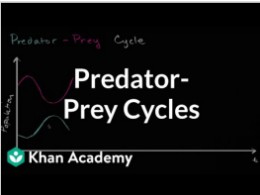

Predator prey cycle | Ecology |
Khan Academy

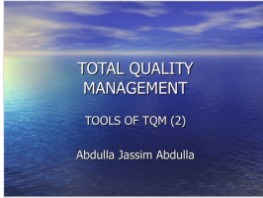

TOTAL QUALITY MANAGEMENT TOOLS OF
TQM (2) Abdulla Jassim Abdulla

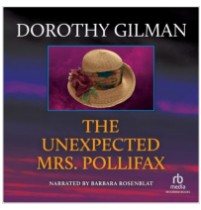

The Unexpected Mrs. Pollifax
Audiobook, by Dorothy Gilman

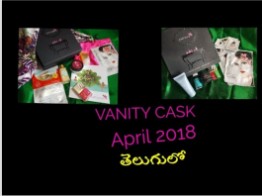

VANITY CASK April 2018  ||
unboxing in Telugu || shades of

Figure 6:  CLIP ViT-B/32 top scoring examples.

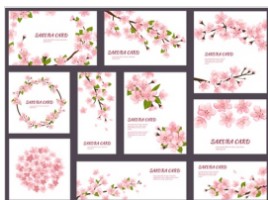

Sakura vector blossom cherry greeting cards with spring pink blooming flowers illustration japanese set of wedding invitation flowering template decoration isolated on white background. Vectores

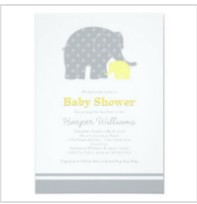

Elephant Baby Shower Invitations | Yellow & Gray

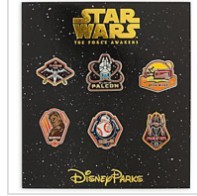

Star Wars: The Force Awakens Pin Trading Booster Set - Disney Parks 2016

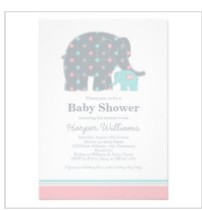

elephant baby shower invitations pink blue grey

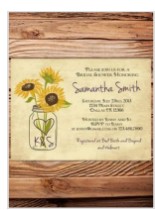

Sunflowers Bridal Shower Invitation,Rustic Sunflowers,Vintage Mason Jar Invitation,Gray, Brown, Mason Jar, Sunflower, Wedding Shower, 5145

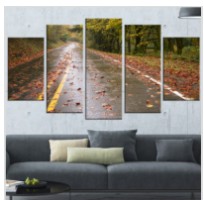

Wet Rainy Road in Forest Landscape Photo Canvas Art Print - 5 Panels

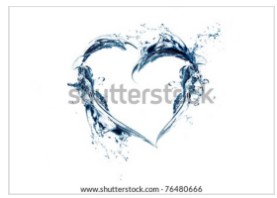

heart shape made of water splash - stock photo

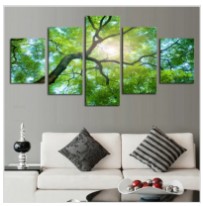

Green Nature Abstract Background Canvas Print Painting Home Decor Wall Art 5Pcs

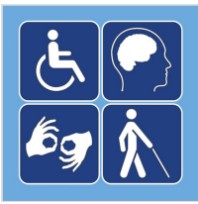

disabled person: Vector set of disability symbols in blue and white Illustration

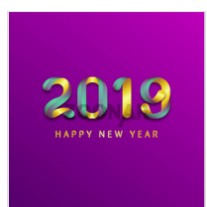

Inscription Happy new year 2019 on purple background

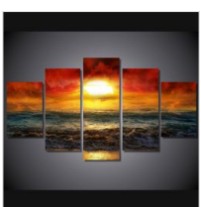

Wholesale Original Oil Painting Framed - 5 Pcs Set No Framed HD Printed amazing sunset artistic Painting on canvas room decoration print poster picture canvas original paintings

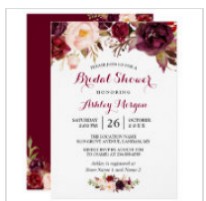

Burgundy Marsala Red Floral Autumn Bridal Shower Invitations

Figure 7: DFN-FT top scoring examples.

