# OpenReview forum: "Filter Like You Test: Data-Driven Data Filtering for CLIP Pretraining"
_NeurIPS.cc/2025/Conference — NeurIPS 2025 poster_

### Official Review · Reviewer_oBzJ · 2025-06-01

**Clarity:** 2
**Significance:** 2
**Originality:** 2
**Rating:** 4
**Confidence:** 2

**Summary:**

This paper presents a data filtering approach for improving CLIP pre-training by automatically identifying high-quality text-image pairs. The authors develop a learned scoring function that predicts how valuable each data point will be for CLIP model training. They implement this approach in two variants - M-FLYT and E-FLYT - which use different feature extraction techniques. Beyond scoring, they introduce SCS, a method that determines the optimal frequency for including each data point in the final training dataset. When evaluated on the DataComp benchmark, their approach outperforms all competing methods that rely solely on publicly available resources.

**Questions:**

1. Why optimize both the scoring model and CLIP reference model together? This seems like a non-trivial choice. What happens if you just train the scoring model while keeping the reference model frozen? I'm not convinced that jointly training both models will converge properly - as they could interfere with each other during training. It would be helpful to see a comparison with the simpler approach of fixing the reference model to understand if the joint optimization is actually necessary.

2. What exactly does public-resource mean in the context of DataComp? I don't think it was explained in the paper.

3. How do your embeddings differ from the original CLIP embeddings? In what cases are your embeddings better and worse than the original CLIP embeddings?

4. Is there a qualitative boost in performance in addition to the quantitative gain?

**Ethical Concerns:**

["NO or VERY MINOR ethics concerns only"]

**Final Justification:**

After reading all of the other reviews and the authors' responses to them, I have decided to raise my score to a 4.
Additional analyses, as mentioned in the exchange between myself and the authors, would be required to convince me to raise my score any higher.

**Limitations:**

yes

**Quality:**

2

**Strengths And Weaknesses:**

Strengths:

1. The paper introduces a novel framework for filtering training data for CLIP pre-training.

2. The authors' method demonstrates superior performance compared to other publicly-available approaches on the DataComp benchmark.

Weaknesses:

1. The evaluation lacks qualitative analysis that would clarify the practical implications of the boost in quantitative performance. The paper would benefit from visual demonstrations showing how the new CLIP embeddings differ from those of the original model - for instance, through image retrieval examples or t-sne embedding visualizations.

2. The writing suffers from slight redundancy and could be tightened. Some concepts and findings are reiterated unnecessarily throughout the paper.

---

> ### Author Rebuttal · Authors · 2025-07-31
>
> We thank the reviewer for recognizing the novelty of our method, and for acknowledging our method's strong performance compared to other publicly-available approaches. Below, we address the weaknesses and questions raised in the review.
>
> > The evaluation lacks qualitative analysis that would clarify the practical implications of the boost in quantitative performance. The paper would benefit from visual demonstrations showing how the new CLIP embeddings differ from those of the original model - for instance, through image retrieval examples or t-sne embedding visualizations.
>
> > How do your embeddings differ from the original CLIP embeddings? In what cases are your embeddings better and worse than the original CLIP embeddings?
>
> > Is there a qualitative boost in performance in addition to the quantitative gain?
>
> We appreciate the reviewer's suggestions for qualitative analysis. We note that such embedding-level qualitative evaluation is not standard practice in the data filtering literature for CLIP pretraining. We would appreciate it if the reviewer could point to papers in the data filtering or CLIP pretraining literature that include such qualitative analysis.
>
> As an alternative, we could provide a different form of qualitative analysis: examining which training examples our method prioritizes compared to existing approaches. This could provide insights as to which types of examples our learned scoring model values differently from other methods.
>
> > The writing suffers from slight redundancy and could be tightened. Some concepts and findings are reiterated unnecessarily throughout the paper.
>
> The clarity and readability of the paper is very important to us. We would be grateful if the reviewer could point to specific instances where our writing is redundant or where concepts are unnecessarily reiterated. We are happy to revise our paper to improve clarity and conciseness.
>
> > Why optimize both the scoring model and CLIP reference model together? This seems like a non-trivial choice. What happens if you just train the scoring model while keeping the reference model frozen?
>
> We agree with the reviewer that optimizing the reference CLIP model is a non-trivial choice. In Appendix C.7 we present E-FLYT ablation experiments including freezing the reference CLIP model. The results are shown in Table 11 under the "Trainable parameters" experiment, "Only scoring model" variation. The experiment shows that training the reference model yields considerably better performance than freezing it.
>
> Intuitively, when the reference model is frozen at a single point in parameter space, the scoring model may overfit to a data weighting that is only locally beneficial for training. In contrast, when we train the reference model, the loss function for the scoring model averages over many configurations in parameter space.

---

> > ### Comment · Reviewer_oBzJ · 2025-08-02
> >
> > I thank the authors for their rebuttal
> >
> > >We appreciate the reviewer's suggestions for qualitative analysis. We note that such embedding-level qualitative evaluation is not standard practice in the data filtering literature for CLIP pretraining. We would appreciate it if the reviewer could point to papers in the data filtering or CLIP pretraining literature that include such qualitative analysis.
> > As an alternative, we could provide a different form of qualitative analysis: examining which training examples our method prioritizes compared to existing approaches. This could provide insights as to which types of examples our learned scoring model values differently from other methods.
> >
> > I agree with the authors that their suggested form of qualitative analysis would be interesting, and it may help them draw conclusions on what makes an example valuable. I also think that some analysis, like the ones I suggested, on the embeddings of the CLIP models trained with different forms of data filtering would also be insightful. As I do not come from the field of data filtering, I am not familiar with what is standard practice, but I do think that my suggested analysis is beneficial regardless.
> >
> > > The clarity and readability of the paper is very important to us. We would be grateful if the reviewer could point to specific instances where our writing is redundant or where concepts are unnecessarily reiterated. We are happy to revise our paper to improve clarity and conciseness.
> >
> > While I respect the authors' choice on how they decided to organize their paper, I do think that it could be organized in a way that reduces redundancies and improves clarity. My issues are mainly with sections 1-3.
> >
> > Section 1 is written as a condensed methods section. I would instead take more time to introduce the problem setting explicitly, which would help readers that are less familiar with dataset filtering or the DataComp benchmark. I would move lines 82-85 from section 2 to section 1 and expand it to include the fact that you are allowed to use additional data for the filtering process and that the quantitative comparisons can also be done against clip models that are trained on other datasets, whether public (CLIP) or non-public (OpenCLIP).
> >
> > Additionally you could remove technical-heavy parts from Section 1 to Section 3. This would reduce the redundancy between the two sections and in turn make the introduction section more high-level.

---

> > > ### Author Response · Authors · 2025-08-04
> > >
> > > Thank you for your reply and useful suggestions!
> > >
> > > **Qualitative analysis**
> > >
> > > We will look into the embedding-level analysis the reviewer suggested and include it in our revision if we find meaningful insights.
> > >
> > > For now, we have begun examining which training examples our method prioritizes compared to existing approaches. Our initial analysis suggests that our method tends to value examples containing _multiple_ ImageNet classes. For instance, our top-scoring example has the text "Honey the Golden Retriever stands in a tent," where both "Golden Retriever" and "tent" correspond to ImageNet classes. In contrast, the highest-scoring examples using the CLIP score filtering method with OpenAI's ViT-B/32 model contain almost exclusively images with visible text that matches the example's text.
> > >
> > > **Writing suggestions**
> > >
> > > We appreciate the reviewer's thoughtful and detailed feedback on the organization of sections 1-3. We particularly like your suggestions about presenting the data filtering problem more explicitly, moving some content from section 2 to section 1.  We will pay close attention to these recommendations as we revise the writing.

---

### Official Review · Reviewer_94Jw · 2025-06-11

**Clarity:** 2
**Significance:** 3
**Originality:** 2
**Rating:** 4
**Confidence:** 3

**Summary:**

This paper proposes Filter Like You Test(FLYT), a data filtering algorithm that learns to assign importance weights to each training examples based on their contribution to the downstream task. This paper also introduce Mixing-FLYT(M-FLYT), which integrates scores from multiple existing filtering methods into a unified score using learned combinations, and Soft Cap Sampling (SCS), a sampling strategy that prevents over-representation of high-scoring examples. Together, these methods achieve state-of-the-art results on the DataComp medium-scale benchmark, surpassing previous public-resource approaches.

**Questions:**

Please address the questions listed in the weaknesses.

**Ethical Concerns:**

["NO or VERY MINOR ethics concerns only"]

**Final Justification:**

I have carefully reviewed the authors' response and acknowledge that it addresses most of my concerns.

**Limitations:**

yes

**Paper Formatting Concerns:**

No Paper Formatting Concerns

**Quality:**

3

**Strengths And Weaknesses:**

Strengths:
1. The paper introduces a learning-based filtering algorithm FLYT, which use directly optimizes the scoring model based on the performance of downstream task. It can further be extend to M-FLYT, where the learnable parameters are the weight of different existing scoring methods.
2. By combining FLYT with Mixing-FLYT and Soft Cap Sampling, the paper indeed achieves state-of-the-art results on the DataComp medium-scale benchmark, including a 5.5% improvement in ImageNet zero-shot accuracy over previous public-resource methods. It proves the effectiveness of the proposed algorithm.

Weaknesses:
1. The Soft Cap Sampling won't eliminate the repetition of sampled data. Could you explain how the proposed Soft Cap Sampling change the distribution of sampled data?
2. The proposed FLYT algorithm appears conceptually and technically aligned with meta-learning paradigms: concept (meta learning: learning to learn) and technique (optimizing the updated parameters by second derivative). However, the paper does not acknowledge this connection or cite relevant meta-learning literature in the introduction or related work.
3. The porposed FLYT/Mixing-FLYT may significantly increases the training time (use second-order gradients to optimize scoring model), which is not discussed in the experiments or mentioned in the limitations section of the paper.

---

> ### Author Rebuttal · Authors · 2025-07-31
>
> We thank the reviewer for recognizing the effectiveness of our direct learning-based approach, especially compared to public-resource approaches. We address the weaknesses raised in the review below.
>
> > The Soft Cap Sampling won't eliminate the repetition of sampled data.
>
> We do not aim to eliminate the repetition of sampled data. Our aim with Soft Cap Sampling is to carefully choose how many times each example is repeated during training.
>
> > Could you explain how the proposed Soft Cap Sampling change the distribution of sampled data?
>
> Figure 2 shows the histogram of the sampled examples using SCS compared to the standard top 20% threshold filtering. We observe that SCS samples more unique examples - 25.7M compared to 23.9M of threshold filtering, while the highest scoring examples are sampled up to 56 times compared to 6 times of threshold filtering. This allows for more diversity in the training distribution while upsampling the best examples.
>
> > The proposed FLYT algorithm appears conceptually and technically aligned with meta-learning paradigms (...) However, the paper does not acknowledge this connection or cite relevant meta-learning literature in the introduction or related work.
>
> In the paragraph "Learned data weighting" of the related work section, we cite [37], [43] which to our knowledge are the two meta-learning papers most relevant to our work and describe their connection with our work.
>
> However, we agree that we should make a clearer connection between our work and meta-learning paradigms in our related work section, and will address this in the final revision. We thank the reviewer for this observation and welcome suggestions for additional especially related papers.
>
> > The porposed FLYT/Mixing-FLYT may significantly increases the training time (use second-order gradients to optimize scoring model), which is not discussed in the experiments or mentioned in the limitations section of the paper.
>
> As noted in Section 4.1, we discuss in detail the computational cost of our method in Appendix C.2. While second-order gradients are generally expensive, our M-FLYT scoring model requires relatively few training iterations, which overall results in 3x fewer FLOPs than training a DataComp medium scale model. At larger scales (DataComp large or Xlarge), filtering cost becomes negligible relative to the overall training cost. In the revision we will mention this under limitations, though we do not consider computational cost a significant limitation.

---

> > ### Comment · Reviewer_94Jw · 2025-08-04
> > **Response to Rebuttal**
> >
> > Thank you for your explanation. I have carefully reviewed your response and acknowledge that it addresses most of my concerns.

---

### Official Review · Reviewer_sYPG · 2025-06-26

**Clarity:** 2
**Significance:** 3
**Originality:** 3
**Rating:** 4
**Confidence:** 3

**Summary:**

The paper proposes a learned sample importance algorithm for filtering a dataset of varying sample quality to one that can effectively train a CLIP model. In the first phase, they train a model by simultaneously learning the CLIP model itself, along with a scoring model that predicts how important each example in the batch is, based on a downstream task loss. The scoring model is of primary interest, as once this phase finishes, they then use it to construct an N (e.g. 128M) example dataset (potentially with duplicates). To allow for duplicates, they propose SoftCap sampling, which smoothly degrades the importance of previously sampled examples, allowing them to be sampled again, but at reduced likelihood. Further variations and improvements are proposed, however the thrust of the idea remains.

**Questions:**

Can you explain how the "Average" column in Table 1 is computed? It's clearly not a simple mean over the former fields. Particularly, it seems as though "M-FLYT+SCS" should have the highest average, given that it beats every other method on the other categories, and yet it has a lower average score than the non-proposed methods above.

**Ethical Concerns:**

["NO or VERY MINOR ethics concerns only"]

**Final Justification:**

The method is effective, and the paper is technically solid. it doesn't require private datasets, and is very light on added heuristics. For these reasons, I think it's suitable to accept the work.

**Limitations:**

yes

**Quality:**

3

**Strengths And Weaknesses:**

### Strengths

* The method is minimal with heuristics, instead relying on the learned scoring model.
* The proposed model does exceedingly well, beating contemporary methods not only on the downstream task (ImageNet), but also other tasks.
* The proposed method doesn't rely on private datasets

### Weaknesses

* Not levied at this paper alone, but the method does directly target ImageNet by using it as the downstream task, so generalization appears to rely on the generalizability of the concepts in ImageNet. This is where the concurrently proposed MGD approach, utilizing all tasks, may result in more robust vision foundation models (of which CLIP is typically considered), since it's not overfitting to ImageNet.
* The clarity of the paper could be improved. It would perhaps be nice to read about FLYT first, to understand the algorithm, before additional complexities such as M-FLYT and E-FLYT are introduced. For example, section 3.5 could appear before 3.3 and 3.4.

---

> ### Author Rebuttal · Authors · 2025-07-31
>
> We thank the reviewer for recognizing what we consider one of the core strengths of our work - minimal reliance on heuristics. We appreciate their acknowledgment of our method's strong performance across multiple tasks, without relying on private datasets. Below, we address the weaknesses and questions raised in the review.
>
> > Not levied at this paper alone, but the method does directly target ImageNet by using it as the downstream task, so generalization appears to rely on the generalizability of the concepts in ImageNet. This is where the concurrently proposed MGD approach, utilizing all tasks, may result in more robust vision foundation models (of which CLIP is typically considered), since it's not overfitting to ImageNet.
>
> Our method can be easily applied using any downstream task. In fact, in Appendix C.9 we experiment with E-FLYT using 22 downstream tasks training sets (similar to MGD). Our initial experiments (Table 12) show no significant difference in performance when using all 22 tasks compared to only using ImageNet.
>
> We chose to focus on ImageNet since it represents standard practice in the field, as the reviewer notes. More importantly, an argument is usually made in the opposite direction: that using all available training sets of the evaluation downstream tasks leads to overfitting on the evaluation sets, while only using ImageNet as a proxy demonstrates better generalization by showing that we are less dependent on the specific downstream task.
>
> > The clarity of the paper could be improved. It would perhaps be nice to read about FLYT first, to understand the algorithm, before additional complexities such as M-FLYT and E-FLYT are introduced. For example, section 3.5 could appear before 3.3 and 3.4.
>
> We thank the reviewer for the suggestion on clarity, as it is very important to us, and an outside perspective is extremely useful. While we consider Sections 3.3, 3.4, and 3.5 to be implementation choices, we agree with the reviewer that Section 3.5 (Weighted CLIP loss) is more crucial for understanding FLYT since it introduces a new loss function. In our final revision, we will move this to section 3.3. We welcome any additional suggestions or feedback the reviewer may have regarding clarity.
>
> > Can you explain how the "Average" column in Table 1 is computed? ... Particularly, it seems as though "M-FLYT+SCS" should have the highest average, given that it beats every other method on the other categories, and yet it has a lower average score than the non-proposed methods above.
>
> The “Average” column represents the average over all 38 downstream tasks of the DataComp benchmark. Of these 38 datasets, 16 are not represented in any of the other 4 categories. These categories follow the standard evaluation protocol for filtering methods in the DataComp benchmark:
>
> * **ImageNet**: ImageNet classification only
> * **ImageNet distribution shifts**: 6 datasets including ImageNet-V2, ImageNet-O
> * **VTAB** (Visual Task Adaptation Benchmark): 13 datasets including Caltech101, CIFAR100
> * **Retrieval**: 3 retrieval datasets - Flickr30k, MSCOCO, WinoGAViL
>
> As a concrete case, on the Camelyon17 dataset, which is only part of the average calculation, our method scores 56.8% while another method (HYPE+DFN+s-CLIPLoss) scores 69.2%. That explains how our average score can be 0.1% lower despite outperforming in all other categories.

---

> > ### Comment · Reviewer_sYPG · 2025-08-04
> >
> > I thank the authors for their response. It addresses my concerns.

---

### Official Review · Reviewer_mNXk · 2025-06-30

**Clarity:** 3
**Significance:** 3
**Originality:** 3
**Rating:** 5
**Confidence:** 3

**Summary:**

This work proposes a method to filter / curate large-scale image-caption datasets for training CLIP-like models. The method consists of two components:
1. A learnable scoring function (M-FLYT) that estimates how useful a data example is for training. It takes as input scores from 12 existing scoring methods and learns to a linear combination of them. The scoring function is learned using a meta-learning strategy that learns the scoring function weights as as to improve the zero-shot classification performance (on ImageNet) of a reference CLIP model.
2. A sampling approach (SCS) that selects training examples based on the learned scores (from M-FLYT). It samples with replacement but adds a penalty $\alpha$ each time an example is selected. This prevents extensive repetitions of examples with high scores.

The method and its individual components are evalauted on the DataComp medium-scale benchmark. It outperforms previous works, achieving 40.1% accuracy on ImageNet zero-shot classification and 37.7% on average across DataComp tasks.

**Questions:**

Please answer my comments and questions put under the "Weakness" section above.

**Ethical Concerns:**

["NO or VERY MINOR ethics concerns only"]

**Final Justification:**

The rebuttal generally resolve my concerns. While the improvements from the proposed methodology are modest compared to some strong baselines, they appear consistent and worthwhile. After also reading the other reviews and authors' responses to them, I maintain my positive assessment and recommend acceptance.

**Limitations:**

Yes

**Paper Formatting Concerns:**

No.

**Quality:**

3

**Strengths And Weaknesses:**

**Strengths:**
- The paper is overall easy to understand and well-written
- The proposed meta-learning approach is conceptually simple (but hard to implement) and the idea of meta-learning the scoring function so as to explicitly improve the downstream task performance makes sense. Furthermore, the proposed sampling approach SCS is simple, easy to implement, and effective.
- The work demonstrates that both M-FLYT approach and the SCS sampling approach improves the results compared to the baselines.

**Weaknesses:**
Although the proposed meta-learning-based FLYT approach is conceptually simple, implementing such meta-optimization techniques is complex (see, for example, the discussion in Appendix A), while the gains in Tables 1 and 2 appear modest. M-FLYT shows relatively small improvement over "DFN-FT [our reproduction of 16]" (compared to the impact of SCS). Moreover, the performance gap between M-FLYT, which mixes individual scores using a computationally heavy and complicated meta-optimization stage, and the simpler, training-free "IN-weighted" method (Table 2, right) is even smaller.

Table 1 suggests that the SCS sampling strategy has a much larger impact on performance than M-FLYT. It is likely that using SCS with just the "IN-weighted" baseline (or even DFN-FT) would yield strong results, close to "M-FLYT + SCS", while being far simpler. Could the authors provide results without M-FLYT or E-FLYT to confirm this? Such results are necessary to judge the practical usefulness of FLYT.

---

> ### Author Rebuttal · Authors · 2025-07-31
>
> We thank the reviewer for finding our paper well-written and easy to understand, for recognizing the soundness of our meta-learning approach, and for acknowledging the effectiveness of our proposed methods. Below, we address the weaknesses raised in the review.
>
>  > implementing such meta-optimization techniques is complex
>
> We agree that implementing FLYT is not trivial, but we think that this makes our implementation (which we open source) interesting to the community.
>
> > (...) M-FLYT, which mixes individual scores using a computationally heavy and complicated meta-optimization stage
>
> First, we note that the meta-optimization stage occurs only during the scoring model training, which we run for just 20M examples compared to the 128M used in DataComp medium scale. As detailed in Appendix C.2, the entire M-FLYT training process (the meta-optimization stage) requires 3x fewer FLOPs than training a DataComp medium scale model. At larger scales (DataComp large or Xlarge) this becomes negligible relative to the overall training cost. Moreover, our learned M-FLYT model is a linear model, which means scoring each example is very fast and computationally inexpensive.
>
> > (...) the gains in Tables 1 and 2 appear modest. M-FLYT shows relatively small improvement over "DFN-FT [our reproduction of 16]" (compared to the impact of SCS). Moreover, the performance gap between M-FLYT (…) and the simpler, training-free "IN-weighted" method (Table 2, right) is even smaller.
>
> We invested heavily in strengthening our baselines: “ImageNet-weighted” is not a naive implementation. Specifically, the method is (as detailed in Appendix C.5):
>
> * Standardize the per-example scores produced by each method
> * Normalize the ImageNet accuracies of these methods to [0,1]
> * Add $1/(r - 1)$ to these normalized accuracies to achieve a desired min to max ratio $r$
> * Compute the weighted sum per example
>
> ImageNet-weighted also requires tuning the hyper parameter $r$. It outperforms more naive baselines such as standardized sum, yet M-FLYT still outperforms it. While we agree the improvement is modest, it is nonetheless clear. When taking into account our previous point that M-FLYT is in fact not computationally expensive, as well as our next point about the built-in calibration of M-FLYT scores, we believe these improvements make M-FLYT worthwhile.
>
> > It is likely that using SCS with just the "IN-weighted" baseline (or even DFN-FT) would yield strong results
>
> A key advantage of the FLYT training algorithm is that it naturally produces a distribution over the training examples. Other methods such as DFN-FT or IN-weighted provide scores that are not easily translated to probability distributions.
>
> That said, we agree with the reviewer that this claim should be supported by experimental results. We ran an experiment of SCS with a repetition penalty $\alpha = 0.15$ on the scores produced by IN-weighted with $r=8$, converting the scores to a probability distribution by applying softmax (following the same approach as with M-FLYT), and present our results in the table below. We will include these results in our revised paper.
>
> For easy comparison, the table also includes IN-weighted with $r=8$ without SCS, in addition to M-FLYT with and without SCS.
>
> | Experiment | ImageNet | Average |
> | :--------- | :---: | :---: |
> | IN-weighted + SCS | 0.379 | 0.362 |
> | IN-weighted | 0.353 | 0.365 |
> | M-FLYT + SCS | 0.401 | 0.377 |
> | M-FLYT | 0.359 | 0.371 |
>
> We believe these results could be improved by introducing a new scale hyperparameter where the distribution becomes softmax(scale * scores). However, this would be significantly more computationally demanding, requiring hyperparameter sweeps over both the new scale parameter and the repetition penalty of SCS.

---

> > ### Comment · Reviewer_mNXk · 2025-08-04
> > **Response to rebuttal**
> >
> > I thank the authors for answering my comments.  The responses generally resolve my concerns. While the improvements are modest compared to some strong baselines, they appear consistent and worthwhile. After also reading the other reviews and authors' responses to them, I maintain my positive assessment and recommend acceptance. I will finalize my score after the discussion with reviewers and the AC.

---

### Comment · Area_Chair_Hds6 · 2025-08-02
**Author-reviewer discussion**

Dear Authors and Reviewers,

I would like to thank the authors for providing detailed rebuttal messages.

For the **reviewers**, I would like to encourage you to carefully read all other reviews and the author responses and engage in an open exchange with the authors. Please post your first response as soon as possible within the discussion time window, so there is time for back and forth discussion with the authors. All reviewers should respond to the authors, so that the authors know their rebuttal has been read.

Cheers,
AC

---

### Decision · Program_Chairs · 2025-09-17

**Decision:**

Accept (poster)

**Comment:**

The paper introduces a data filtering algorithm to learn a scoring function (M-FLYT) that estimates the usefulness of each data example for training a CLIP model. In the algorithm, a CLIP model is trained along with a scoring model that predicts importance of each example in a batch based on downstream task losses. The scoring model takes 12 existing filtering methods as input and learns to a linear combination with meta-learning strategy. Techniques such as Soft Cap Sampling (SCS) are leveraged to refine the sampling (e.g., preventing over-representation of high-scoring examples and smoothly reducing the importance of previously sampled examples). Improvement is showed on DataComp medium-scale benchmark and ImageNet zero-shot classification tasks.


Strength of the paper:
- Well written
- Conceptually simple method. A novel framework to leverage meta-learning in data filtering
- Good selection of downstream evaluation tasks supports the efficacy of proposed approach. Improvements are modest compared to some strong baselines, but appear consistent.
- Working on a popular and important area (vision language representation learning) and providing a empirically effective algorithm to the area

Reviewers align on most of the strengths of the paper from the high level and see its novelty, contribution, and relevance to NeurIPS venue. Thus we are inclined to accept this paper to the conference. However, there are a few areas  that can be improved, after most questions being addressed and clarified during the rebuttal and discussion stage

To improve:

- Although the proposed meta-learning-based FLYT approach is conceptually simple, implementing such meta-optimization techniques is complex. Leveraging downstream task or filters to select examples might also introduce uncontrolled biases and reduce model generalizability.
- Would be great to see the method being applied to different variants of CLIP (e.g., SigLIP, MetaCLIP, etc.)
- Many discussion between reviewers and authors are valuable to include in the final manuscript